# Provably Consistent Partial-Label Learning

**Lei Feng**[1]* **Jiaqi Lv**[2] **Bo Han**[3] **Miao Xu**[4,5]
**Gang Niu**[5] **Xin Geng**[2] **Bo An**[1]† **Masashi Sugiyama**[5,6]

[1]School of Computer Science and Engineering, Nanyang Technological University, Singapore
[2]School of Computer Science and Engineering, Southeast University, Nanjing, China
[3]Department of Computer Science, Hong Kong Baptist University, China
[4]The University of Queensland, Australia
[5]Center for Advanced Intelligence Project, RIKEN, Japan
[6]The University of Tokyo, Japan

## Abstract

*Partial-label learning* (PLL) is a multi-class classification problem, where each training example is associated with *a set of candidate labels*. Even though many practical PLL methods have been proposed in the last two decades, there lacks a theoretical understanding of the consistency of those methods—none of the PLL methods hitherto possesses a *generation process* of candidate label sets, and then it is still unclear why such a method works on a specific dataset and when it may fail given a different dataset. In this paper, we propose the first *generation model* of candidate label sets, and develop two novel PLL methods that are guaranteed to be provably consistent, i.e., one is *risk-consistent* and the other is *classifier-consistent*. Our methods are advantageous, since they are compatible with any deep network or stochastic optimizer. Furthermore, thanks to the generation model, we would be able to answer the two questions above by testing if the generation model matches given candidate label sets. Experiments on benchmark and real-world datasets validate the effectiveness of the proposed generation model and two PLL methods.

## 1 Introduction

Unlike supervised and unsupervised learning, *weakly supervised learning* [72] aims to learn under weak supervision. So far, various weakly supervised learning frameworks have been widely studied. Examples include semi-supervised learning [7, 4, 55, 59, 58, 46, 51, 5, 39], multi-instance learning [1, 73], positive-unlabeled learning [14, 15, 54, 29, 22, 9], complementary-label learning [32, 68, 33, 64, 11], noisy-label learning [52, 49, 57, 27, 67, 50, 44, 60, 65, 25, 61], positive-confidence learning [34], similar-unlabeled learning [2], and unlabeled-unlabeled learning [42, 43].

This paper focuses on learning under another natural type of weak supervision, which is called *partial-label learning* (PLL) [35, 13, 40, 10, 70, 18, 48]. PLL aims to deal with the problem where each instance is provided with a set of candidate labels, only one of which is the correct label. In some studies, PLL is also termed as *ambiguous-label learning* [31, 69, 10, 8, 66] and *superset-label learning* [41, 40, 21]. Due to the difficulty in collecting accurately labeled data in many real-world scenarios, PLL has been successfully applied to a wide range of application domains, such as web mining [45], bird song classification [41], and automatic face naming [69].

A number of methods [35, 53, 70, 17, 18] have been proposed to improve the practical performance of PLL; on the theoretical side, some researchers have studied the statistical consistency [13] and the

learnability [40] of PLL. They made the same assumption on the *ambiguity degree*, which describes the maximum co-occurring probability of the correct label with another candidate label. Although they assumed that the data distribution for successful PLL should ensure a limited ambiguity degree, it is still unclear what the explicit formulation of the data distribution would be. Besides, the consistency of PLL methods would be hardly guaranteed without modeling the data distribution.

Motivated by the above observations, we for the first time present a novel statistical model to depict the *generation process of candidate label sets*. Having an explicit data distribution not only helps us to understand how partially labeled examples are generated, but also enables us to perform empirical risk minimization. We verify that the proposed generation model satisfies the key assumption of PLL that the correct label is always included in the candidate label set. Based on the generation model, we have the following contributions:

- We derive a novel *risk-consistent* method and a novel *classifier-consistent* method. Most of the existing PLL methods need to specially design complex optimization objectives, which make the optimization process inefficient. In contrast, our proposed PLL methods are model-independent and optimizer-independent, and thus can be naturally applied to complex models such as deep neural networks with any advanced optimizer.

- We derive an estimation error bound for each of the two methods, which demonstrates that the obtained empirical risk minimizer would approximately converge to the true risk minimizer as the number of training data tends to infinity. We show that the risk-consistent method holds a tighter estimation error bound than the classifier-consistent method and empirically validate that the risk-consistent method achieves better performance when deep neural networks are used.

- To show the effect of our generation model, we also use *entropy* to measure how well the given candidate label sets match our generation model. We find that the candidate label sets with higher entropy better match our generation model, and on such datasets, our proposed PLL methods achieve better performance.

Extensive experiments on benchmark as well as real-world partially labeled datasets clearly validate the effectiveness of our proposed methods.

## 2 Formulations

In this section, we introduce some notations and briefly review the formulations of learning with ordinary labels, learning with partial labels, and learning with complementary labels.

**Learning with Ordinary Labels.** For ordinary multi-class learning, let the feature space be $\mathcal{X} \in \mathbb{R}^d$ and the label space be $\mathcal{Y} = [k]$ (with $k$ classes) where $[k] := \{1, 2, \ldots, k\}$. Let us clearly define that $\boldsymbol{x}$ denotes an instance and $(\boldsymbol{x}, y)$ denotes an example including an instance $\boldsymbol{x}$ and a label $y$. When ordinary labels are provided, we usually assume each example $(\boldsymbol{x}, y) \in \mathcal{X} \times \mathcal{Y}$ is independently sampled from an unknown data distribution with probability density $p(\boldsymbol{x}, y)$. Then, the goal of multi-class learning is to obtain a multi-class classifier $f : \mathcal{X} \to \mathbb{R}^k$ that minimizes the following classification risk:

$$R(f) = \mathbb{E}_{p(\boldsymbol{x}, y)}[\mathcal{L}(f(\boldsymbol{x}), y)], \tag{1}$$

where $\mathbb{E}_{p(\boldsymbol{x}, y)}[\cdot]$ denotes the expectation over the joint probability density $p(\boldsymbol{x}, y)$ and $\mathcal{L} : \mathbb{R}^k \times \mathcal{Y} \to \mathbb{R}_+$ is a multi-class loss function that measures how well a classifier estimates a given label. We say that a method is *classifier-consistent* if the learned classifier by the method is infinite-sample consistent to $\arg\min_{f \in \mathcal{F}} R(f)$, and a method is *risk-consistent* if the method possesses a classification risk estimator that is equivalent to $R(f)$ given the same classifier $f$. Note that a risk-consistent method is also classifier-consistent [62]. However, a classifier-consistent method may not be risk-consistent.

**Learning with Partial Labels.** For learning with partial labels (i.e., PLL), each instance is provided with a set of candidate (partial) labels, only one of which is correct. Suppose the partially labeled dataset is denoted by $\widetilde{\mathcal{D}} = \{(\boldsymbol{x}_i, Y_i)\}_{i=1}^n$ where $Y_i$ is the candidate label set of $\boldsymbol{x}_i$. Since each candidate label set should not be the empty set nor the whole label set, we have $Y_i \in \mathcal{C}$ where $\mathcal{C} = \{2^{\mathcal{Y}} \setminus \emptyset \setminus \mathcal{Y}\}$, $2^{\mathcal{Y}}$ denotes the power set, and $|\mathcal{C}| = 2^k - 2$. The key assumption of PLL lies in that the correct label $y_i$ of $\boldsymbol{x}_i$ must be in the candidate label set, i.e.,

$$p(y_i \in Y_i \mid \boldsymbol{x}_i, Y_i) = 1, \ \forall (\boldsymbol{x}_i, y_i) \in \mathcal{X} \times \mathcal{Y}, \ \forall Y_i \in \mathcal{C}. \tag{2}$$

Given such data, the goal of PLL is to induce a multi-class classifier $f : \mathcal{X} \rightarrow \mathbb{R}^k$ that can make correct predictions on test inputs. To this end, many methods [41, 70, 71, 21, 18, 48] have been proposed to improve the performance of PLL. However, to the best of our knowledge, there is only one method [13] that possesses statistical consistency by providing a classifier-consistent risk estimator. However, it not only requires the assumption that the data distribution should ensure a limited ambiguity degree, but also relies on some strict conditions (e.g., convexity of loss function and dominance relation [13]). It is still unclear what the explicit formulation of the data distribution for successful PLL would be. Besides, it is also unknown whether there exists a risk-consistent method that possesses a statistical unbiased estimator of the classification risk $R(f)$.

**Learning with Complementary Labels.** There is a special case of partial labels, called complementary labels [32, 68, 33]. Each complementary label specifies one of the classes that the example does *not* belong to. Hence a complementary label $\overline{y}$ can be considered as an extreme case where all $k - 1$ classes other than the class $\overline{y}$ are taken as candidate (partial) labels. Existing studies on learning with complementary labels make the assumption on the data generation process. The pioneering study [32] assumed that each complementarily labeled example $(\boldsymbol{x}, \overline{y})$ is independently drawn from the probability distribution with density $\overline{p}(\boldsymbol{x}, y)$, where $\overline{p}(\boldsymbol{x}, y)$ is defined as $\overline{p}(\boldsymbol{x}, \overline{y}) = \sum_{y \neq \overline{y}} p(\boldsymbol{x}, y)$. Based on this data distribution, several risk-consistent methods [32, 33] have been proposed for learning with complementary labels. However, in many real-world scenarios, multiple complementary labels would be more widespread than a single complementary label. Hence a recent study [19] focused on learning with multiple complementary labels. Suppose each training example is represented by $(\boldsymbol{x}, \overline{Y})$ where $\overline{Y}$ denotes a set of multiple complementary labels, and $(\boldsymbol{x}, \overline{Y})$ is assumed to be independently sampled from the probability distribution with density $\overline{p}(\boldsymbol{x}, \overline{Y})$, which is defined as

$$\overline{p}(\boldsymbol{x}, \overline{Y}) = \sum_{j=1}^{k-1} p(s = j)\overline{p}(\boldsymbol{x}, \overline{Y} \mid s = j), \tag{3}$$

where

$$\overline{p}(\boldsymbol{x}, \overline{Y} \mid s = j) := \begin{cases} \frac{1}{\binom{k-1}{j}} \sum_{y \notin \overline{Y}} p(\boldsymbol{x}, y) & \text{if } |\overline{Y}| = j, \\ 0 & \text{otherwise.} \end{cases} \tag{4}$$

Here, the variable $s$ denotes the size of the complementary label set. Supplied with this data distribution, a risk-consistent method [19] was proposed. It is worth noting that following the distribution of complementarily labeled data, although we can obtain partial labels by regarding all the complementary labels as non-candidate labels, the resulting distribution of partially labeled data is not explicitly formulated. It would be natural to ask whether there also exists an explicit formulation of the partially labeled data distribution that enables us to derive a novel classifier-consistent method or a novel risk-consistent method that possesses statistical consistency. In this paper, we will give an affirmative answer to this question. Specifically, we will show that based on our proposed data generation model, a novel risk-consistent method (the first one for PLL) and a novel classifier-consistent method can be derived accordingly.

## 3 Data Generation Model

### 3.1 Partially Labeled Data Distribution

We assume each partially labeled example $(\boldsymbol{x}, Y)$ is independently drawn from a probability distribution with the following density:

$$\widetilde{p}(\boldsymbol{x}, Y) = \sum_{i=1}^{k} p(Y \mid y = i)p(\boldsymbol{x}, y = i), \text{ where } p(Y \mid y = i) = \begin{cases} \frac{1}{2^{k-1} - 1} & \text{if } i \in Y, \\ 0 & \text{if } i \notin Y. \end{cases} \tag{5}$$

In Eq. (5), we assume $p(Y \mid \boldsymbol{x}, y) = p(Y \mid y)$, which means, given the correct label $y$, the candidate label set $Y$ is independent of the instance $\boldsymbol{x}$. This assumption is similar to the conventional modeling of label noise [26] where the observed noisy label is independent of the instance, given the correct label. In addition, there are in total $2^{k-1} - 1$ possible candidate label sets that contain a specific label $y$. Hence, Eq. (5) describes the probability of each candidate label set being uniformly sampled, given a specific label. Here, we show that our assumed data distribution is a valid probability distribution by the following theorem.

**Theorem 1.** *The equality $\int_{\mathcal{C}} \int_{\mathcal{X}} \widetilde{p}(\boldsymbol{x}, Y)\mathrm{d}\boldsymbol{x} \, \mathrm{d}Y = 1$ holds.*

The proof is provided in Appendix A.1. Given the assumed data distribution in Eq. (5), it would be natural to ask whether our assumed data distribution meets the key assumption of PLL described in Eq. (2), i.e., whether the correct label $y$ is always in the candidate label set $Y$ for every partially labeled example $(\boldsymbol{x}, Y)$ sampled from $\widetilde{p}(\boldsymbol{x}, Y)$. The following theorem provides an affirmative answer to this question.

**Theorem 2.** *For any partially labeled example $(\boldsymbol{x}, Y)$ independently sampled from the assumed data distribution in Eq. (5), the correct label $y$ is always in the candidate label set $Y$, i.e., $p(y \in Y \mid \boldsymbol{x}, Y) = 1$, $\forall (\boldsymbol{x}, Y) \sim \widetilde{p}(\boldsymbol{x}, Y)$.*

The proof is provided in Appendix A.2. Theorem 2 clearly demonstrates that our assumed data distribution in Eq. (5) satisfies the key assumption of PLL.

## 3.2 Motivation

Here, we provide a motivation why we derived the above data generation model. Generally, a large number of high-quality samples are notably helpful to machine learning or data mining. However, it is usually difficult for our labelers to directly identify the correct label for each instance [72]. Nonetheless, it would be easier to collect a set of candidate labels that contains the correct label. Suppose there is a labeling system that can uniformly sample a label set $Y$ from $\mathcal{C}$. For each instance $\boldsymbol{x}$, the labeling system uniformly samples a label set $Y$ and asks a labeler whether the correct label $y$ is in the sampled label set $Y$. In this case, the collected examples whose correct label $y$ is included in the proposed label set $Y$ follow the same distribution as Eq. (5). In order to justify that, we first introduce the following lemma.

**Lemma 1.** *Given any instance $\boldsymbol{x}$ with its correct label $y$, for any unknown label set $Y$ that is uniformly sampled from $\mathcal{C}$, the equality $p(y \in Y \mid \boldsymbol{x}) = 1/2$ holds.*

It is quite intuitive to verify that Lemma 1 indeed holds. Specifically, if we do not have any information of $Y$, we may randomly guess with even probabilities whether the correct $y$ is included in an unknown label set $Y$ or not. A rigorous mathematical proof is provided in Appendix A.3. Based on Lemma 1, we have the following theorem.

**Theorem 3.** *In the above setting, the distribution of the collected data whose correct label $y \in \mathcal{Y}$ is included in the label set $Y \in \mathcal{C}$ is the same as Eq. (5), i.e., $p(\boldsymbol{x}, Y \mid y \in Y) = \widetilde{p}(\boldsymbol{x}, Y)$ where $\widetilde{p}(\boldsymbol{x}, Y)$ is defined in Eq. (5).*

The proof is provided in Appendix A.4.

## 4 Consistent Methods

In this section, based on our assumed partially labeled data distribution in Eq. (5), we present a novel risk-consistent method and a novel classifier-consistent method and theoretically derive an estimator error bound for each of them. Both methods are agnostic in specific classification models and can be easily trained with stochastic optimization, which ensures their scalability to large-scale datasets.

### 4.1 Risk-Consistent Method

For the risk-consistent method, we employ the *importance reweighting* strategy [23] to rewrite the classification risk $R(f)$ as

$$
\begin{aligned}
R(f) &= \mathbb{E}_{p(\boldsymbol{x}, y)}[\mathcal{L}(f(\boldsymbol{x}), y)] = \int_{\boldsymbol{x}} \sum_{i=1}^{k} p(y = i \mid \boldsymbol{x}) \mathcal{L}(f(\boldsymbol{x}), i) p(\boldsymbol{x}) \mathrm{d}\boldsymbol{x} \\
&= \int_{\boldsymbol{x}} \sum_{i=1}^{k} \frac{1}{|\mathcal{C}|} \sum_{Y \in \mathcal{C}} p(Y \mid \boldsymbol{x}) \frac{p(y = i \mid \boldsymbol{x})}{p(Y \mid \boldsymbol{x})} \mathcal{L}(f(\boldsymbol{x}), i) p(\boldsymbol{x}) \mathrm{d}\boldsymbol{x} \\
&= \frac{1}{|\mathcal{C}|} \int_{\boldsymbol{x}} \sum_{Y \in \mathcal{C}} p(Y \mid \boldsymbol{x}) \Big[ \sum_{i=1}^{k} \frac{p(y = i \mid \boldsymbol{x})}{p(Y \mid \boldsymbol{x})} \mathcal{L}(f(\boldsymbol{x}), i) \Big] p(\boldsymbol{x}) \mathrm{d}\boldsymbol{x} \\
&= \frac{1}{2^{k} - 2} \mathbb{E}_{\widetilde{p}(\boldsymbol{x}, Y)} \Big[ \sum_{i=1}^{k} \frac{p(y = i \mid \boldsymbol{x})}{p(Y \mid \boldsymbol{x})} \mathcal{L}(f(\boldsymbol{x}), i) \Big] = R_{\mathrm{rc}}(f).
\end{aligned}
\tag{6}
$$

Here, $p(Y \mid \boldsymbol{x})$ can be calculated by

$$
p(Y \mid \boldsymbol{x}) = \sum_{j=1}^{k} p(Y \mid y = j) p(y = j \mid \boldsymbol{x}) = \frac{1}{2^{k-1} - 1} \sum_{j \in Y} p(y = j \mid \boldsymbol{x}),
\tag{7}
$$

where the last equality holds due to Eq. (5). By substituting Eq. (7) into Eq. (6), we obtain

$$R_{\mathrm{rc}}(f) = \frac{1}{2}\mathbb{E}_{\widetilde{p}(\boldsymbol{x},Y)}\left[\sum_{i=1}^{k}\frac{p(y=i|\boldsymbol{x})}{\sum_{j\in Y}p(y=j|\boldsymbol{x})}\mathcal{L}\big(f(\boldsymbol{x}),i\big)\right]. \tag{8}$$

In this way, its empirical risk estimator can be expressed as

$$\widehat{R}_{\mathrm{rc}}(f) = \frac{1}{2n}\sum_{o=1}^{n}\left(\sum_{i=1}^{k}\frac{p(y_o=i|\boldsymbol{x}_o)}{\sum_{j\in Y_o}p(y_o=j|\boldsymbol{x}_o)}\mathcal{L}\big(f(\boldsymbol{x}_o),i\big)\right), \tag{9}$$

where $\{\boldsymbol{x}_o, Y_o\}_{o=1}^{n}$ are partially labeled examples drawn from $\widetilde{p}(\boldsymbol{x}, Y)$. Note that $p(y = i \mid \boldsymbol{x})$ is not accessible from the given data. Therefore, we apply the softmax function on the model output $f(\boldsymbol{x})$ to approximate $p(y = i \mid \boldsymbol{x})$, i.e., $p(y = i \mid \boldsymbol{x}) = g_i(\boldsymbol{x})$ where $g_i(\boldsymbol{x})$ is the probability of label $i$ being the true label of $\boldsymbol{x}$, which is calculated by $g_i(\boldsymbol{x}) = \exp(f_i(\boldsymbol{x}))/\sum_{j=1}^{k}\exp(f_j(\boldsymbol{x}))$, and $f_i(\boldsymbol{x})$ is the $i$-th coordinate of $f(\boldsymbol{x})$. Note that the non-candidate labels can never be the correct label. Hence we further correct $p(y = i \mid \boldsymbol{x})$ by setting the confidence of each non-candidate label to 0, i.e.,

$$p(y = i \mid \boldsymbol{x}) = g_i(\boldsymbol{x}) \text{ if } i \in Y, \text{ otherwise } p(y = i \mid \boldsymbol{x}) = 0, \ \forall(\boldsymbol{x}, Y) \sim \widetilde{p}(\boldsymbol{x}, Y). \tag{10}$$

As shown in Eq. (9), our risk-consistent method does not rely on specific loss functions, hence we simply adopt the widely-used categorical cross entropy loss for practical implementation. The pseudo-code of the Risk-Consistent (RC) method is presented in Algorithm 1. It is worth noting that the algorithmic process of RC surprisingly coincides with that of PRODEN [47]. However, they are derived in totally different manners. Besides, PRODEN does not hold any theoretical guarantee while we show that our proposed RC method is consistent.

Here, we establish an estimation error bound for our RC method to demonstrate its learning consistency. Let $\widehat{f}_{\mathrm{rc}} = \min_{f\in\mathcal{F}}\widehat{R}_{\mathrm{rc}}(f)$ be the empirical risk minimizer and $f^{\star} = \min_{f\in\mathcal{F}}R(f)$ be the true risk minimizer. Besides, we define the function space $\mathcal{H}_y$ for the label $y \in \mathcal{Y}$ as $\{h : \boldsymbol{x} \mapsto f_y(\boldsymbol{x}) \mid f \in \mathcal{F}\}$. Let $\mathfrak{R}_n(\mathcal{H}_y)$ be the expected Rademacher complexity [3] of $\mathcal{H}_y$ with sample size $n$, then we have the following theorem.

**Theorem 4.** *Assume the loss function $\mathcal{L}(f(\boldsymbol{x}), y)$ is $\rho$-Lipschitz with respect to $f(\boldsymbol{x})$ $(0 < \rho < \infty)$ for all $y \in \mathcal{Y}$ and upper-bounded by $M$, i.e., $M = \sup_{\boldsymbol{x}\in\mathcal{X}, f\in\mathcal{F}, y\in\mathcal{Y}}\mathcal{L}(f(\boldsymbol{x}), y)$. Then, for any $\delta > 0$, with probability at least $1 - \delta$,*

$$R(\widehat{f}_{\mathrm{rc}}) - R(f^{\star}) \le 4\sqrt{2}\rho\sum_{y=1}^{k}\mathfrak{R}_n(\mathcal{H}_y) + M\sqrt{\frac{\log\frac{2}{\delta}}{2n}},$$

The proof of Theorem 4 is provided in Appendix B. Generally, $\mathfrak{R}_n(\mathcal{H}_y)$ can be bounded by $C_{\mathcal{H}}/\sqrt{n}$ for a positive constant $C_{\mathcal{H}}$ [43, 62, 20]. Hence Theorem 4 shows that the empirical risk minimizer $f_{\mathrm{rc}}$ converges to the true risk minimizer $f^{\star}$ as $n \to \infty$.

## 4.2 Classifier-Consistent Method

For the classifier-consistent method, we start by introducing a transition matrix $\boldsymbol{Q}$ that describes the probability of the candidate label set given an ordinary label. Specifically, the transition matrix $\boldsymbol{Q}$ is defined as $Q_{ij} = p(Y = C_j \mid y = i)$ where $C_j \in \mathcal{C}$ $(j \in [2^k - 2])$ is a specific label set. By further taking into account the assumed data distribution in Eq. (5), we can instantiate the transition matrix $\boldsymbol{Q}$ as $Q_{ij} = \frac{1}{2^{k-1}-1}$ if $i \in C_j$, otherwise $Q_{ij} = 0$. Let us introduce $q_j(\boldsymbol{x}) = p(Y = C_j \mid \boldsymbol{x})$ and $g_i(\boldsymbol{x}) = p(y = i \mid \boldsymbol{x})$, then we can obtain $q(\boldsymbol{x}) = \boldsymbol{Q}^{\top}g(\boldsymbol{x})$ with the assumption $p(Y \mid \boldsymbol{x}, y) = p(Y \mid y)$. Given each partially labeled example $(\boldsymbol{x}, Y)$ sampled from $\widetilde{p}(\boldsymbol{x}, Y)$, the proposed classifier-consistent risk estimator is presented as

$$R_{\mathrm{cc}}(f) = \mathbb{E}_{\widetilde{p}(\boldsymbol{x},Y)}[\mathcal{L}(q(\boldsymbol{x}), \widetilde{y})], \text{ where } Y = C_{\widetilde{y}}. \tag{11}$$

In this formulation, we regard the candidate label set $Y$ as a virtual label $\widetilde{y}$ if $Y$ is a specific label set $C_{\widetilde{y}}$. Since there are $2^k - 2$ possible label sets, we denote by $\widetilde{\mathcal{Y}}$ the virtual label space where $\widetilde{\mathcal{Y}} = [2^k - 2]$ and $\widetilde{y} \in \widetilde{\mathcal{Y}}$. It is worth noting that the transition matrix $\boldsymbol{Q}$ has full rank, because all rows of $\boldsymbol{Q}$ are linearly independent by the definition of $\boldsymbol{Q}$. Then, in order to prove that this method is classifier-consistent, we introduce the following lemma.

**Lemma 2.** *If certain loss functions are used (e.g., the softmax cross entropy loss or mean squared error), by minimizing the expected risk $R(f)$, the optimal mapping $g^{\star}$ satisfies $g_i^{\star}(\boldsymbol{x}) = p(y = i \mid \boldsymbol{x})$.*

The proof is provided in Appendix C.1. The same proof can also be found in [68, 47].

| **Algorithm 1** RC Algorithm | **Algorithm 2** CC Algorithm |
|---|---|
| **Input:** Model $f$, epoch $T_{\max}$, iteration $I_{\max}$, partially labeled training set $\mathcal{D} = \{(\boldsymbol{x}_i, Y_i)\}_{i=1}^n$. | **Input:** Model $f$, epoch $T_{\max}$, iteration $I_{\max}$, partially labeled training set $\mathcal{D} = \{(\boldsymbol{x}_i, Y_i)\}_{i=1}^n$; |
| 1: **Initialize** $p(y_i = j \mid \boldsymbol{x}_i) = 1, \forall j \in Y_i$, otherwise $p(y_i = j \mid \boldsymbol{x}_i) = 0$; | 1: **for** $t = 1, 2, \ldots, T_{\max}$ **do** |
| 2: **for** $t = 1, 2, \ldots, T_{\max}$ **do** | 2:    **Shuffle** the partially labeled training set $\widetilde{\mathcal{D}} = \{(\boldsymbol{x}_i, Y_i)\}_{i=1}^n$; |
| 3:    **Shuffle** $\widetilde{\mathcal{D}} = \{(\boldsymbol{x}_i, Y_i)\}_{i=1}^n$; | 3:     **for** $j = 1, \ldots, I_{\max}$ **do** |
| 4:     **for** $j = 1, \ldots, I_{\max}$ **do** | 4:       **Fetch** mini-batch $\widetilde{\mathcal{D}}_j$ from $\widetilde{\mathcal{D}}$; |
| 5:       **Fetch** mini-batch $\widetilde{\mathcal{D}}_j$ from $\widetilde{\mathcal{D}}$; | 5:       **Update** model $f$ by minimizing the empirical risk estimator $\widehat{R}_{\mathrm{cc}}$ in Eq. (12); |
| 6:       **Update** model $f$ by $\widehat{R}_{\mathrm{rc}}$ in Eq. (9); | 6:     **end for** |
| 7:       **Update** $p(y_i \mid \boldsymbol{x}_i)$ by Eq. (10); | 7: **end for** |
| 8:     **end for** | **Output:** $f$. |
| 9: **end for**    **Output:** $f$. | |

**Theorem 5.** *When the transition matrix $\boldsymbol{Q}$ has full rank and the condition in Lemma 2 is satisfied, the minimizer $f_{\mathrm{cc}} = \arg\min_{f \in \mathcal{F}} R_{\mathrm{cc}}(f)$ is also the true minimizer $f^\star = \arg\min_{f \in \mathcal{F}} R(f)$, i.e., $f_{\mathrm{cc}} = f^\star$ (classifier-consistency).*

The proof is provided in Appendix C.2.

As suggested by Lemma 2, we adopt the cross entropy loss in our classifier-consistent risk estimator (i.e., Eq. (11)) for practical implementation. In this way, we have the following empirical risk estimator:

$$\widehat{R}_{\mathrm{cc}}(f) = -\frac{1}{n}\sum_{i=1}^n \Big(\sum_{j=1}^{2^k-2}\mathbb{I}(Y_i = C_j)\log(q_j(\boldsymbol{x}_i))\Big) = -\frac{1}{n}\sum_{i=1}^n\sum_{j=1}^{2^k-2}\mathbb{I}(Y_i = C_j)\log\big(\boldsymbol{Q}[:,j]^\top g(\boldsymbol{x})\big)$$

$$= -\frac{1}{n}\sum_{i=1}^n\log\Big(\frac{1}{2^{k-1}-1}\sum_{y\in Y_i}g_y(\boldsymbol{x})\Big) = -\frac{1}{n}\sum_{i=1}^n\log\Big(\frac{1}{2^{k-1}-1}\sum_{y\in Y_i}\frac{\exp(f_y(\boldsymbol{x}))}{\sum_j \exp(f_j(\boldsymbol{x}))}\Big), \qquad (12)$$

where $\mathbb{I}[\cdot]$ is the indicator function. For the expected risk estimator $R_{\mathrm{cc}}(f)$, it seems that the transition matrix $\boldsymbol{Q} \in \mathbb{R}^{k \times (2^k - 2)}$ is indispensable. Unfortunately, it would be computationally prohibitive, since $2^k - 2$ is an extremely large number if the number of classes $k$ is large. However, for practical implementation, Eq. (12) shows that we do not need to explicitly calculate and store the transition matrix $\boldsymbol{Q}$, which brings no pain to optimization. The pseudo-code of the Classifier-Consistent (CC) method is presented in Algorithm 2.

Here, we also establish an estimation error bound for the classifier-consistent method. Let $\widehat{f}_{\mathrm{cc}} = \arg\min_{f \in \mathcal{F}} \widehat{R}_{\mathrm{cc}}(f)$ be the empirical minimizer and $f^\star = \arg\min_{f \in \mathcal{F}} R(f)$ be the true minimizer. Besides, we define the function space $\mathcal{H}_y$ for the label $y \in \mathcal{Y}$ as $\{h : \boldsymbol{x} \mapsto f_y(\boldsymbol{x}) \mid f \in \mathcal{F}\}$. Then, we have the following theorem.

**Theorem 6.** *Assume the loss function $\mathcal{L}(q(\boldsymbol{x}), \widetilde{y})$ is $\rho'$-Lipschitz with respect to $f(\boldsymbol{x})$ $(0 < \rho < \infty)$ for all $\widetilde{y} \in \widetilde{\mathcal{Y}}$ and upper-bounded by $M$, i.e., $M = \sup_{\boldsymbol{x} \in \mathcal{X}, f \in \mathcal{F}, \widetilde{y} \in \widetilde{y}} \mathcal{L}(q(\boldsymbol{x}), \widetilde{y})$. Then, for any $\delta > 0$, with probability at least $1 - \delta$,*

$$R_{\mathrm{cc}}(\widehat{f}_{cc}) - R_{\mathrm{cc}}(f^\star) \le 4\sqrt{2}\rho' \sum_{y=1}^k \mathfrak{R}_n(\mathcal{H}_y) + 2M\sqrt{\frac{\log\frac{2}{\delta}}{2n}}.$$

The proof is provided in Appendix D. Theorem 6 demonstrates that the empirical risk minimizer $\widehat{f}_{\mathrm{cc}}$ converges to the true risk minimizer $f^\star$ as $n \to \infty$.

**Theoretical Comparison Between RC and CC.** There exists a clear difference between the estimation error bounds in Theorem 4 and Theorem 6, especially in the last term. If we assume that $\rho$ for RC and $\rho'$ for CC hold the same value, we can find that the estimation error bound in Theorem 6 would be looser than that in Theorem 4. Therefore, we could expect that RC may have better performance than CC. In addition, RC needs to estimate the prediction confidence of each example. Intuitively, complex models like deep neural networks normally provide more accurate estimation than linear models. Therefore, we speculate that when more complex models are used, the superiority of RC would be more remarkable. We will demonstrate via experiments that RC is generally superior to CC when deep neural networks are used.

Table 1: Test performance (mean±std) of each method using neural networks on benchmark datasets. ResNet is trained on CIFAR-10, and MLP is trained on the other three datasets.

| | MNIST | Kuzushiji-MNIST | Fashion-MNIST | CIFAR-10 |
|---|---|---|---|---|
| RC | **98.00±0.11%** | **89.38±0.28%** | **88.38±0.16%** | **77.93±0.59%** |
| CC | 97.87±0.10%• | 88.83±0.40%• | 87.88±0.25%• | 75.78±0.27%• |
| GA | 96.37±0.13%• | 84.23±0.19%• | 85.57±0.16%• | 72.22±0.19%• |
| NN | 96.75±0.08%• | 82.36±0.41%• | 86.25±0.14%• | 68.09±0.31%• |
| Free | 88.48±0.37%• | 70.31±0.68%• | 81.34±0.47%• | 17.74±1.20%• |
| PC | 92.47±0.13%• | 73.45±0.20%• | 83.37±0.31%• | 46.53±2.01%• |
| Forward | 97.64±0.11%• | 87.64±0.13%• | 86.73±0.15%• | 71.18±0.92%• |
| EXP | 97.81±0.04%• | 88.48±0.29%• | 87.96±0.06%• | 73.22±0.66%• |
| LOG | 97.86±0.11%• | 88.24±0.08%• | 88.31±0.26% | 75.38±0.34%• |
| MAE | 97.82±0.11%• | 88.43±0.32%• | 87.83±0.22%• | 66.91±3.08%• |
| MSE | 96.95±0.14%• | 85.16±0.44%• | 85.72±0.26%• | 66.15±2.13%• |
| GCE | 96.71±0.08%• | 85.19±0.39%• | 86.88±0.16%• | 68.39±0.71%• |
| Phuber-CE | 95.10±0.34%• | 80.66±0.41%• | 85.33±0.23%• | 58.60±0.95%• |

Table 2: Test performance (mean±std) of each method using neural networks on benchmark datasets. DenseNet is trained on CIFAR-10, and LeNet is trained on the other three datasets.

| | MNIST | Kuzushiji-MNIST | Fashion-MNIST | CIFAR-10 |
|---|---|---|---|---|
| RC | **99.04±0.03%** | **94.00±0.30%** | **89.48±0.15%** | **78.53±0.46%** |
| CC | 98.99±0.08% | 93.86±0.18% | 88.98±0.20%• | 75.71±0.18%• |
| GA | 98.68±0.05%• | 90.39±0.26%• | 87.95±0.12%• | 71.85±0.19%• |
| NN | 98.51±0.08%• | 89.60±0.34%• | 88.47±0.15%• | 71.98±0.35%• |
| Free | 80.48±2.06%• | 71.18±1.38%• | 74.02±3.88%• | 45.94±0.83%• |
| PC | 95.03±0.16%• | 79.62±0.11%• | 83.98±0.20%• | 54.18±2.10%• |
| Forward | 98.80±0.04%• | 93.87±0.14% | 88.72±0.17%• | 73.56±1.47%• |
| EXP | 98.82±0.03%• | 92.69±0.31%• | 88.99±0.25%• | 75.02±1.02%• |
| LOG | 98.88±0.08%• | 93.97±0.25% | 88.75±0.28%• | 75.54±0.59%• |
| MAE | 98.88±0.05%• | 93.04±0.52%• | 87.30±3.16%• | 67.74±0.89%• |
| MSE | 98.38±0.05%• | 88.37±0.55%• | 88.18±0.08%• | 70.66±0.59%• |
| GCE | 98.63±0.06%• | 91.27±0.30%• | 88.66±0.16%• | 72.09±0.51%• |
| Phuber-CE | 96.92±0.18%• | 82.24±2.45%• | 87.02±0.09%• | 66.47±0.35%• |

Table 3: Test performance (mean±std) of each method using linear model on UCI datasets.

| | Texture | Yeast | Dermatology | Har | 20Newsgroups |
|---|---|---|---|---|---|
| RC | 99.24±0.14% | 59.89±1.27% | 99.41±1.00% | 98.03±0.09% | **75.99±0.53%** |
| CC | 98.02±2.91%• | **59.97±1.57%** | **99.73±0.85%** | **98.10±0.18%** | 75.97±0.54% |
| SURE | 95.38±0.28%• | 54.39±1.32%• | 97.48±0.32%• | 97.43±0.24%• | 69.82±0.26%• |
| CLPL | 91.93±0.97%• | 54.58±2.11%• | 99.62±0.85% | 97.48±0.18%• | 71.44±0.55%• |
| PLECOC | 69.69±4.82%• | 37.37±9.73%• | 87.84±5.30%• | 96.97±0.29%• | 15.32±7.86%• |
| PLSVM | 49.38±9.99%• | 45.70±8.01%• | 80.00±7.53%• | 91.64±1.43%• | 32.59±8.91%• |
| PLKNN | 96.78±0.31%• | 47.79±2.41%• | 80.54±5.06%• | 94.17±0.59%• | 27.18±0.65%• |
| IPAL | **99.45±0.23%** | 48.99±3.84%• | 98.65±2.27%• | 96.55±0.40%• | 48.36±0.85%• |

# 5 Experiments

In this section, we conduct extensive experiments on various datasets to validate the effectiveness of our proposed methods.

**Datasets.** We collect four widely used benchmark datasets including MNIST [38], Kuzushiji-MNIST [12], Fashion-MNIST [63], and CIFAR-10 [37], and five datasets from the UCI Machine Learning Repository [37]. In order to generate candidate label sets on these datasets, following the motivation in Section 3.2, we uniformly sample the candidate label set that includes the correct label from $\mathcal{C}$ for each instance. In addition, we also use five widely used real-world partially labeled datasets, including Lost [13], BirdSong [6], MSRCv2 [41], Soccer Player [69], Yahoo! News [24]. Since our proposed methods do not rely on specific classification models, we use various base models to validate the effectiveness of our methods, including linear model, three-layer ($d$-500-$k$) MLP, 5-layer LeNet, 34-layer ResNet [28], and 22-layer DenseNet [30]. The detailed descriptions of these datasets with the corresponding base models are provided in Appendix E.1.

**Compared Methods.** We compare with six state-of-the-art PLL methods including SURE [18], CLPL [13], IPAL [70], PLSVM [16], PLECOC [71], PLKNN [31]. Besides, we also compare with various *complementary-label learning* (CLL) methods for two reasons: 1) We can directly use CLL

Table 4: Test performance (mean±std) of each method using linear model on real-world datasets.

|  | Lost | MSRCv2 | BirdSong | Soccer Player | Yahoo! News |
|---|---|---|---|---|---|
| RC | **79.43±3.26%** | 46.56±2.71% | 71.94±1.72% | **57.00±0.97%** | **68.23±0.83%** |
| CC | 79.29±3.19% | 47.22±3.02% | **72.22±1.71%** | 56.32±0.64% | 68.14±0.81% |
| SURE | 71.33±3.57%● | 46.88±4.67% | 58.92±1.28%● | 49.41±086%● | 45.49±1.15%● |
| CLPL | 74.87±4.30%● | 36.53±4.59%● | 63.56±1.40%● | 36.82±1.04%● | 46.21±0.90%● |
| PLECOC | 49.03±8.36%● | 41.53±3.25%● | 71.58±1.81% | 53.70±2.02%● | 66.22±1.01%● |
| PLSVM | 75.31±3.81%● | 35.85±4.41%● | 49.90±2.07%● | 46.29±0.96%● | 56.85±0.91%● |
| PLKNN | 36.73±2.99%● | 41.36±2.89%● | 64.94±1.42%● | 49.62±0.67%● | 41.07±1.02%● |
| IPAL | 72.12±4.48%● | **50.80±4.46%○** | 72.06±1.55% | 55.03±0.77%● | 66.79±1.22%● |

Table 5: Test performance (mean±std) of the RC method using neural networks on benchmark datasets with different generation models.

|  | Case 1 | Case 2 | Case 3 | Case 4 | Case 5 | Our Case | Supervised |
|---|---|---|---|---|---|---|---|
| MLP MNIST | 95.29● (±0.14) | 97.17● (±0.04) | 97.68● (±0.10) | 97.93 (±0.15) | 98.97 (±0.12) | **98.00** (±0.11) | 98.48 (±0.00) |
| MLP KMNIST | 79.88● (±0.47) | 85.65● (±0.38) | 88.04● (±0.37) | 89.07● (±0.20) | 89.34 (±0.18) | **89.38** (±0.21) | 91.53 (±0.00) |
| MLP FMNIST | 79.78● (±0.32) | 84.97● (±0.32) | 87.05● (±0.17) | 88.09● (±0.18) | 88.27 (±0.24) | **88.38** (±0.23) | 89.37 (±0.00) |
| LeNet MNIST | 98.82● (±0.05) | 99.02 (±0.06) | 99.02 (±0.06) | **99.04** (±0.08) | **99.04** (±0.05) | **99.04** (±0.08) | 99.22 (±0.00) |
| LeNet KMNIST | 92.81● (±0.39) | 93.54● (±0.21) | 93.71● (±0.20) | 93.77● (±0.23) | 93.89 (±0.25) | **94.00** (±0.31) | 95.34 (±0.00) |
| LeNet FMNIST | 81.59● (±0.18) | 86.49● (±0.31) | 88.48● (±0.15) | 89.24● (±0.11) | 89.45 (±0.18) | **89.48** (±0.11) | 89.93 (±0.00) |

Table 6: Test performance (mean±std) of the CC method using neural networks on benchmark datasets with different generation models.

|  | Case 1 | Case 2 | Case 3 | Case 4 | Case 5 | Our Case | Supervised |
|---|---|---|---|---|---|---|---|
| MLP MNIST | 96.36● (±0.17) | 97.49● (±0.10) | 97.76 (±0.12) | 97.85 (±0.08) | **97.87** (±0.17) | **97.87** (±0.10) | 98.48 (±0.00) |
| MLP KMNIST | 80.65● (±0.86) | 86.43● (±0.80) | 88.06● (±0.57) | 88.69 (±0.21) | 88.73 (±0.44) | **88.83** (±0.40) | 91.53 (±0.00) |
| MLP FMNIST | 79.81● (±0.45) | 84.49● (±0.33) | 86.47● (±0.14) | 87.52● (±0.15) | 87.64 (±0.18) | **87.80** (±0.25) | 89.37 (±0.00) |
| LeNet MNIST | 98.28● (±0.19) | 98.83● (±0.08) | 98.93 (±0.07) | 98.94 (±0.02) | 98.95 (±0.09) | **98.99** (±0.08) | 99.22 (±0.00) |
| LeNet KMNIST | 86.67● (±1.22) | 92.16● (±0.30) | 93.13● (±0.26) | 93.41● (±0.30) | 93.81 (±0.22) | **93.86** (±0.18) | 95.34 (±0.00) |
| LeNet FMNIST | 77.75● (±5.32) | 86.11● (±0.31) | 87.86● (±0.20) | 88.53● (±0.31) | 88.97 (±0.25) | **88.98** (±0.20) | 89.93 (±0.00) |

methods on partially labeled datasets by regarding non-candidate labels as complementary labels. 2) Existing CLL methods can be applied to large-scale datasets. The compared CLL methods include GA, NN, and Free [33], PC [32], Forward [68], the unbiased risk estimator [19] with bounded losses MAE, MSE, GCE, Phuber-CE, and the surrogate losses EXP and LOG. For all the above methods, their hyper-parameters are specified or searched according to the suggested parameter settings by respective papers. The detailed information of these compared methods is provided in Appendix E.2. For our proposed methods RC (Algorithm 1) and CC (Algorithm 2), we only need to search learning rate and weight decay from $\{10^{-6}, \ldots, 10^{-1}\}$, since there are no other hyper-parameters in our methods. Hyper-parameters are selected so as to maximize the accuracy on a validation set (10% of the training set) of partially labeled data. We implement them using PyTorch [56] and use the Adam [36] optimizer with the mini-batch size set to 256 and the number of epochs set to 250. For all the parametric methods, we adopt the same base model for fair comparisons.

**Experimental Results.** We run 5 trials on the four benchmark datasets and run 10 trials (with 90%/10% train/test split) on UCI datasets and real-world partially labeled datasets, and record the mean accuracy with standard deviation (mean±std). We also use paired $t$-test at 5% significance level, and ●/○ represents whether the *best* of RC and CC is significantly better/worse than other

Table 7: Test performance (mean±std) of each method using neural networks on benchmark datasets. DenseNet is trained on CIFAR-10, and LeNet is trained on the other three datasets. Candidate label sets are generated by the generation model in Case 1 (entropy=2.015).

|  | MNIST | Kuzushiji-MNIST | Fashion-MNIST | CIFAR-10 |
|---|---|---|---|---|
| RC | **98.82±0.05%** | **92.81±0.39%** | **81.59±0.18%** | **68.18±0.60%** |
| CC | 98.28±0.19%• | 86.67±1.22%• | 77.75±5.32%• | 56.13±3.33%• |
| GA | 97.29±0.19%• | 83.79±0.98%• | 70.91±0.99%• | 41.57±1.35%• |
| NN | 69.51±2.06%• | 51.03±1.88%• | 53.13±2.04%• | 31.54±1.65%• |
| Free | 15.29±0.58%• | 13.60±0.37%• | 10.58±0.54%• | 12.53±0.34%• |
| PC | 96.56±0.25%• | 85.60±0.45%• | 80.98±0.44% | 65.97±0.39%• |
| Forward | 95.87±4.82%• | 90.83±0.82%• | 59.66±2.75%• | 51.25±0.49%• |
| EXP | 84.37±9.30%• | 71.10±5.74%• | 59.56±8.43%• | 30.35±0.38%• |
| LOG | 98.17±0.10%• | 87.85±0.82%• | 77.50±5.12%• | 54.61±4.04%• |
| MAE | 56.81±8.36%• | 49.78±9.03%• | 36.41±0.29%• | 30.61±0.43%• |
| MSE | 95.80±0.24%• | 74.95±0.84%• | 58.85±3.52%• | 58.18±1.25%• |
| GCE | 95.92±0.09%• | 80.49±1.10%• | 72.25±0.35%• | 57.47±0.59%• |
| Phuber-CE | 79.41±1.61%• | 59.88±1.06%• | 58.65±1.22%• | 57.53±3.36%• |

compared methods. Besides, the best results are highlighted in bold. Table 1 and Table 2 report the test performance of each method using neural networks on benchmark datasets. We also provide the transductive performance of each method in Appendix E.3. From the two tables, we can observe that RC always achieves the best performance and significantly outperforms other compared methods in most cases. In addition, we record the test accuracy at each training epoch to provide more detailed visualized results in Appendix E.4. Table 3 and Table 4 report the test performance of each method using linear model on UCI datasets and real-world partially labeled datasets, respectively. We can find that RC and CC generally achieve superior performance against other compared methods on both UCI datasets and real-world partially labeled datasets.

**Performance Comparison Between RC and CC.** It can be seen that when linear model is used, RC and CC achieve similar performance. However, RC significantly outperforms CC when deep neural networks are used. These observations clearly accord with our conjecture that the superiority of RC would be more remarkable when more complex models are used.

**Effectiveness of Generation Model.** Here, we test the performance of our methods under different data generation processes. We use *entropy* to measure how well given candidate label sets match the proposed generation model. By this measure, we could know ahead of model training whether to apply our proposed methods or not on a specific dataset. We expect that the higher the entropy, the better the match, thus the better the performance of our proposed methods. To verify our conjecture, we generate various candidate labels sets by six different cases of generation models, and each of them holds a value of entropy. The detailed information of the six cases is provided in Appendix F. Table 5 and Table 6 report the test performance (mean±std) of the RC method and the CC method using neural networks on benchmark datasets with different cases of generation models. From the two tables, we can observe that the higher the entropy, the better the match, thus the better the performance of our proposed methods. Thus, our conjecture is clearly validated. We further conduct experiments with the generation model of Case 1 where given candidate label sets do not match our proposed generation model well. The experimental results are shown in Table 7. As can be seen from Table 7, our methods still significantly outperform other compared methods and RC always achieves the best performance.

## 6 Conclusion

In this paper, we for the first time provided an explicit mathematical formulation of the partially labeled data generation process for PLL. Based on our data generation model, we further derived a novel *risk-consistent* method and a novel *classifier-consistent* method. To the best of our knowledge, we provided the first risk-consistent PLL method. Besides, our proposed methods do not reply on specific models and can be easily trained with stochastic optimization, which ensures their scalability to large-scale datasets. In addition, we theoretically derived an *estimation error bound* for each of the proposed methods. Finally, extensive experimental results clearly demonstrated the effectiveness of the proposed generation model and two PLL methods.

## Broader Impact

A potential application of our proposed partial-label learning methods would be data privacy. For example, when we collect some survey data, we may ask respondents to answer some extremely private questions. It would be difficult for us to directly obtain the ground-truth answer (label) to the question. However, it would be easier for us to obtain a set of candidate labels that contains the true label, since it is mentally less demanding for respondents to remove several obviously wrong labels. In this case, our proposed partial-label learning methods can be used.

There may also exist some negative impacts of our proposed methods. For example, an adversary might deliberately ask a person to give some candidate choices or remove some improper choices to specially designed questions, so that high-quality partially labeled data could be collected. The adversary may apply the proposed partial-label learning methods to learn from the collected partially labeled data. As a consequence, some extremely private data of the person would be divulged or leveraged by the adversary. In addition, if partial-label learning methods are very effective and prevalent, the need for accurately annotated data would be significantly reduced. As a result, the rate of unemployment for data annotation specialists might be increased.

## Acknowledgements

This research was supported by the National Research Foundation, Singapore under its AI Singapore Programme (AISG Award No: AISG-RP-2019-0013), National Satellite of Excellence in Trustworthy Software Systems (Award No: NSOE-TSS2019-01), and NTU. Any opinions, findings and conclusions or recommendations expressed in this material are those of the author(s) and do not reflect the views of National Research Foundation, Singapore. JL and XG were supported by NSFC (62076063). BH was supported by the RGC Early Career Scheme No. 22200720, NSFC Young Scientists Fund No. 62006202, HKBU Tier-1 Start-up Grant and HKBU CSD Start-up Grant. GN and MS were supported by JST AIP Acceleration Research Grant Number JPMJCR20U3, Japan.

## Footnotes

*Preliminary work was done during an internship at RIKEN AIP.

†Correspondence to: boan@ntu.edu.sg.

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
