[Supplementary Material]

# Provably Consistent Partial-Label Learning: Supplementary Material

## A  Proofs of Data Generation Process

### A.1  Proof of Theorem 1

From our formulation of the partially labeled data distribution $\widetilde{p}(\boldsymbol{x}, Y)$, we can obtain the simplified expression $\widetilde{p}(\boldsymbol{x}, Y) = \frac{1}{2^{k-1}-1} \sum_{y \in Y} p(\boldsymbol{x}, y)$. Then, we have

$$
\begin{aligned}
\int_{\mathcal{C}} \int_{\mathcal{X}} \widetilde{p}(\boldsymbol{x}, Y) \mathrm{d}\boldsymbol{x} \, \mathrm{d}Y &= \int_{\mathcal{X}} \sum_{Y \in \mathcal{C}} \widetilde{p}(\boldsymbol{x}, Y) \mathrm{d}\boldsymbol{x} \\
&= \frac{1}{2^{k-1}-1} \int_{\mathcal{X}} \sum_{Y \in \mathcal{C}} \sum_{y \in Y} p(\boldsymbol{x}, y) \mathrm{d}\boldsymbol{x} \\
&= \frac{1}{2^{k-1}-1} \int_{\mathcal{X}} \sum_{y=1}^{k} \sum_{Y \in \{Y | Y \in \mathcal{C}, y \in Y\}} p(\boldsymbol{x}, y) \mathrm{d}\boldsymbol{x} \\
&= \frac{1}{2^{k-1}-1} \int_{\mathcal{X}} \sum_{y=1}^{k} (2^{k-1} - 1) p(\boldsymbol{x}, y) \mathrm{d}\boldsymbol{x} \\
&= 1,
\end{aligned}
$$

which concludes the proof of Theorem 1.  □

### A.2  Proof of Theorem 2

It is intuitive to express $p(y \in Y \mid \boldsymbol{x}, Y)$ as

$$
\begin{aligned}
p(y \in Y \mid \boldsymbol{x}, Y) &= 1 - p(y \notin Y \mid \boldsymbol{x}, Y) \\
&= 1 - \sum_{i \notin Y} p(y = i \mid \boldsymbol{x}, Y) \\
&= 1 - \sum_{i \notin Y} \frac{p(Y \mid y = i, \boldsymbol{x}) p(y = i \mid \boldsymbol{x})}{p(Y \mid \boldsymbol{x})} \\
&= 1 - \sum_{i \notin Y} \frac{p(Y \mid y = i) p(y = i \mid \boldsymbol{x})}{\sum_{j=1}^{k} p(Y \mid y = j) p(y = j \mid \boldsymbol{x})} \\
&= 1 - (2^{k-1} - 1) \sum_{i \notin Y} \frac{p(Y \mid y = i) p(y = i \mid \boldsymbol{x})}{\sum_{j \in Y} p(y = j \mid \boldsymbol{x})} \\
&= 1,
\end{aligned}
$$

where the last equality holds because $p(Y \mid y = i) = 0$ if $i \notin Y$, in terms of Eq. (5).  □

### A.3 Proof of Lemma 1

Let us first consider the case where the correct label $y$ is a specific label $i$ $(i \in [k])$, then we have

$$
\begin{aligned}
p(y \in Y, y = i \mid \boldsymbol{x}) &= p(y \in Y \mid y = i, \boldsymbol{x})p(y = i \mid \boldsymbol{x}) \\
&= \sum_{C \in \mathcal{C}} p(y \in Y, Y = C \mid y = i, \boldsymbol{x})p(y = i \mid \boldsymbol{x}) \\
&= \sum_{C \in \mathcal{C}} p(y \in Y \mid Y = C, y = i, \boldsymbol{x})p(y = i \mid \boldsymbol{x})p(Y = C \mid \boldsymbol{x}) \\
&= \sum_{C \in \mathcal{C}} p(y \in Y \mid Y = C, y = i, \boldsymbol{x})p(y = i \mid \boldsymbol{x})p(Y = C) \\
&= \frac{1}{2^k - 2} \sum_{C \in \mathcal{C}} p(y \in Y \mid Y = C, y = i, \boldsymbol{x})p(y = i \mid \boldsymbol{x}) \\
&= \frac{1}{2^k - 2} |\mathcal{C}^i| \cdot p(y = i \mid \boldsymbol{x}) \\
&= \frac{2^{k-1} - 1}{2^k - 2} p(y = i \mid \boldsymbol{x}) \\
&= \frac{1}{2} p(y = i \mid \boldsymbol{x}),
\end{aligned}
$$

where we have used $p(Y = C \mid \boldsymbol{x}) = p(Y = C) = \frac{1}{2^k-2}$ because $Y$ is sampled from the whole set of label sets uniformly at random. In addition, $\mathcal{C}^i = \{Y \in \mathcal{C} \mid i \in Y\}$ denotes the set of all the label sets that contain $i$, hence we can obtain $|\mathcal{C}^i| = 2^{k-1} - 1$. By further summing up the both side over all possible $i$, we can obtain

$$
\sum_i p(y \in Y, y = i \mid \boldsymbol{x}) = \sum_i \frac{1}{2} p(y = i \mid \boldsymbol{x}) \Rightarrow p(y \in Y \mid \boldsymbol{x}) = \frac{1}{2},
$$

which concludes the proof of Lemma 1. $\qquad\square$

### A.4 Proof of Theorem 3

Let us express $p(Y \mid y \in Y, \boldsymbol{x})$ as

$$
\begin{aligned}
p(Y \mid y \in Y, \boldsymbol{x}) &= \frac{p(y \in Y, Y \mid \boldsymbol{x})}{p(y \in Y \mid \boldsymbol{x})} \\
&= \frac{p(y \in Y \mid Y, \boldsymbol{x})p(Y \mid \boldsymbol{x})}{p(y \in Y \mid \boldsymbol{x})} \\
&= \frac{p(y \in Y \mid Y, \boldsymbol{x})p(Y)}{p(y \in Y \mid \boldsymbol{x})} \\
&= \frac{2}{2^k - 2} p(y \in Y \mid Y, \boldsymbol{x}) \qquad (\because p(y \in Y \mid \boldsymbol{x}) = \frac{1}{2} \text{ and } p(Y) = \frac{1}{2^k - 2}) \\
&= \frac{1}{2^{k-1} - 1} \sum_{y \in Y} p(y \mid \boldsymbol{x}).
\end{aligned}
$$

By further multiplying $p(\boldsymbol{x})$ on both side, we can obtain $p(\boldsymbol{x}, Y \mid y \in Y) = \frac{1}{2^{k-1}-1} \sum_{y \in Y} p(\boldsymbol{x}, y) = \widetilde{p}(\boldsymbol{x}, Y)$ where $\widetilde{p}(\boldsymbol{x}, Y)$ is our presented data distribution for PLL. $\qquad\square$

## B Proofs of Theorem 4

Our proof of the estimation error bound is based on *Rademacher complexity* [1].

**Definition 1** (Redemacher complexity). *Let $Z_1, \ldots, Z_n$ be $n$ i.i.d. random variables drawn from a probability distribution $\mu$, $\mathcal{H} = \{h : \mathcal{Z} \to \mathbb{R}\}$ be a class of measurable functions. Then the expected*

*Rademacher complexity of $\mathcal{H}$ is defined as*

$$\mathfrak{R}_n(\mathcal{H}) = \mathbb{E}_{Z_1,\ldots,Z_n \sim \mu} \mathbb{E}_{\boldsymbol{\sigma}} \left[ \sup_{h \in \mathcal{H}} \frac{1}{n} \sum_{i=1}^{n} \sigma_i h(Z_i) \right],$$

*where $\boldsymbol{\sigma} = (\sigma_1, \ldots, \sigma_n)$ are Rademacher variables taking the value from $\{-1, +1\}$ with even probabilities.*

Before proving Theorem 4, we introduce the following lemmas.

**Lemma 3.** *Let $\widehat{f}$ be the empirical risk minimizer (i.e., $\widehat{f} = \arg\min_{f \in \mathcal{F}} \widehat{R}(f)$) and $f^\star$ be the true risk minimizer (i.e., $f^\star = \arg\min_{f \in \mathcal{F}} R(f)$), then the following inequality holds:*

$$R(\widehat{f}) - R(f^\star) \leq 2 \sup_{f \in \mathcal{F}} |\widehat{R}(f) - R(f)|.$$

*Proof.* It is intuitive to obtain

$$\begin{aligned} R(\widehat{f}) - R(f^\star) &\leq R(\widehat{f}) - \widehat{R}(\widehat{f}) + \widehat{R}(\widehat{f}) - R(f^\star) \\ &\leq R(\widehat{f}) - \widehat{R}(\widehat{f}) + R(\widehat{f}) - R(f^\star) \\ &\leq 2 \sup_{f \in \mathcal{F}} |\widehat{R}(f) - R(f)|, \end{aligned}$$

which completes the proof. The same proof has been provided in [20]. □

Then, we define a function space for our RC method as

$$\mathcal{G}_{\mathrm{rc}} = \left\{ (\boldsymbol{x}, Y) \mapsto \frac{1}{2} \sum_{i=1}^{k} \frac{p(y = i \mid \boldsymbol{x})}{\sum_{j \in Y} p(y = j \mid \boldsymbol{x})} \mathcal{L}(f(\boldsymbol{x}), i) \mid f \in \mathcal{F} \right\},$$

where $(\boldsymbol{x}, Y)$ is randomly sampled from $\widetilde{p}(\boldsymbol{x}, Y)$. Let $\widetilde{\mathfrak{R}}_n(\mathcal{G}_{\mathrm{rc}})$ be the expected Rademacher complexity of $\mathcal{G}_{\mathrm{rc}}$, i.e.,

$$\widetilde{\mathfrak{R}}_n(\mathcal{G}_{\mathrm{rc}}) = \mathbb{E}_{\widetilde{p}(\boldsymbol{x}, Y)} \mathbb{E}_{\boldsymbol{\sigma}} \left[ \sup_{g \in \mathcal{G}_{\mathrm{rc}}} \frac{1}{n} \sum_{i=1}^{n} \sigma_i g(\boldsymbol{x}_i, Y_i) \right].$$

Then we have the following lemma.

**Lemma 4.** *Suppose the loss function $\mathcal{L}$ is bounded by $M$, i.e., $M = \sup_{\boldsymbol{x} \in \mathcal{X}, f \in \mathcal{F}, y \in \mathcal{Y}} \mathcal{L}(f(\boldsymbol{x}), y)$, then for any $\delta > 0$, with probability at least $1 - \delta$,*

$$\sup_{f \in \mathcal{F}} \left| R_{\mathrm{rc}}(f) - \widehat{R}_{\mathrm{rc}}(f) \right| \leq 2 \widetilde{\mathfrak{R}}_n(\mathcal{G}_{\mathrm{rc}}) + \frac{M}{2} \sqrt{\frac{\log \frac{2}{\delta}}{2n}}.$$

*Proof.* In order to prove this lemma, we first show that the one direction $\sup_{f \in \mathcal{F}} R_{\mathrm{rc}}(f) - \widehat{R}_{\mathrm{rc}}(f)$ is bounded with probability at least $1 - \delta/2$, and the other direction can be similarly shown. Suppose an example $(\boldsymbol{x}_i, Y_i)$ is replaced by another arbitrary example $(\boldsymbol{x}_i', Y_i')$, then the change of $\sup_{f \in \mathcal{F}} R_{\mathrm{rc}}(f) - \widehat{R}_{\mathrm{rc}}(f)$ is no greater than $M/(2n)$, since $\mathcal{L}$ is bounded by $M$. By applying *McDiarmid's inequality* [19], for any $\delta > 0$, with probability at least $1 - \delta/2$,

$$\sup_{f \in \mathcal{F}} R_{\mathrm{rc}}(f) - \widehat{R}_{\mathrm{rc}}(f) \leq \mathbb{E} \left[ \sup_{f \in \mathcal{F}} R_{\mathrm{rc}}(f) - \widehat{R}_{\mathrm{rc}}(f) \right] + \frac{M}{2} \sqrt{\frac{\log \frac{2}{\delta}}{2n}}.$$

Using the same trick in [20], we can obtain

$$\mathbb{E} \left[ \sup_{f \in \mathcal{F}} R_{\mathrm{rc}}(f) - \widehat{R}_{\mathrm{rc}}(f) \right] \leq 2 \widetilde{\mathfrak{R}}_n(\mathcal{G}_{\mathrm{rc}}).$$

By further taking into account the other side $\sup_{f \in \mathcal{F}} \widehat{R}_{\mathrm{rc}}(f) - R_{\mathrm{rc}}(f)$, we have for any $\delta > 0$, with probability at least $1 - \delta$,

$$\sup_{f \in \mathcal{F}} \left| R_{\mathrm{rc}}(f) - \widehat{R}_{\mathrm{rc}}(f) \right| \leq 2 \widetilde{\mathfrak{R}}_n(\mathcal{G}_{\mathrm{rc}}) + \frac{M}{2} \sqrt{\frac{\log \frac{2}{\delta}}{2n}},$$

which concludes the proof. $\qquad\square$

Next, we will bound the expected Rademacher complexity of $\mathcal{G}_{\mathrm{rc}}$ (i.e., $\widetilde{\mathfrak{R}}_n(\mathcal{G}_{\mathrm{rc}})$) by the following lemma.

**Lemma 5.** *Assume the loss function $\mathcal{L}(f(\boldsymbol{x}), y)$ is $\rho$-Lipschitz with respect to $f(\boldsymbol{x})$ $(0 < \rho < \infty)$ for all $y \in \mathcal{Y}$. Then, the following inequality holds:*

$$\widetilde{\mathfrak{R}}_n(\mathcal{G}_{\mathrm{rc}}) \leq \sqrt{2} \rho \sum_{y=1}^k \mathfrak{R}_n(\mathcal{H}_y),$$

*where*

$$\mathcal{H}_y = \{ h : \boldsymbol{x} \mapsto f_y(\boldsymbol{x}) \mid f \in \mathcal{F} \},$$

$$\mathfrak{R}_n(\mathcal{H}_y) = \mathbb{E}_{p(\boldsymbol{x})} \mathbb{E}_{\boldsymbol{\sigma}} \left[ \sup_{h \in \mathcal{H}_y} \frac{1}{n} \sum_{i=1}^n h(\boldsymbol{x}_i) \right].$$

*Proof.* First of all, we introduce $p_i(\boldsymbol{x}) = \frac{p(y=i|\boldsymbol{x})}{\sum_{j \in Y} p(y=j|\boldsymbol{x})}$ for each example $(\boldsymbol{x}, Y)$. Thus we have $0 \leq p_i(\boldsymbol{x}) \leq 1, \forall i \in [k]$ and $\sum_{i=1}^k p_i(\boldsymbol{x}) = 1$ since $p_i(\boldsymbol{x}) = 0$ if $i \notin Y$. In this way, we can obtain $\widetilde{\mathfrak{R}}_n(\mathcal{G}_{\mathrm{rc}}) \leq \mathfrak{R}_n(\mathcal{L} \circ \mathcal{F})$ where $\mathcal{L} \circ \mathcal{F}$ denotes $\{ \mathcal{L} \circ f \mid f \in \mathcal{F} \}$. Since $\mathcal{H}_y = \{ h : \boldsymbol{x} \mapsto f_y(\boldsymbol{x}) \mid f \in \mathcal{F} \}$ and the loss function $\mathcal{L}(f(\boldsymbol{x}), y)$ is $\rho$-Lipschitz with respect to $f(\boldsymbol{x})$ $(0 < \rho < \infty)$ for all $y \in \mathcal{Y}$, by the Rademacher vector contraction inequality [18], we have $\mathfrak{R}_n(\mathcal{L} \circ \mathcal{F}) \leq \sqrt{2} \rho \sum_{y=1}^k \mathfrak{R}_n(\mathcal{H}_y)$, which concludes the proof of Lemma 5. $\qquad\square$

Combining Lemma 3, Lemma 4, and Lemma 5, Theorem 4 is proved. $\qquad\square$

## C  Proofs of Classifier-Consistency

### C.1  Proof of Lemma 2

**Cross Entropy Loss**  If the cross entropy loss is used, we have the following optimization problem:

$$\phi(g) = -\sum_{i=1}^k p(y = i \mid \boldsymbol{x}) \log(g_i(\boldsymbol{x}))$$

$$\text{s.t. } \sum_{i=1}^k g_i(\boldsymbol{x}) = 1.$$

By using the Lagrange multiplier method, we can obtain the following non-constrained optimization problem:

$$\Phi(g) = -\sum_{i=1}^k p(y = i \mid \boldsymbol{x}) \log(g_i(\boldsymbol{x})) + \lambda(\sum_{i=1}^k g_i(\boldsymbol{x}) - 1)).$$

By setting the derivative to 0, we obtain

$$g_i^\star(\boldsymbol{x}) = \frac{1}{\lambda} p(y = i \mid \boldsymbol{x}).$$

Because $\sum_{i=1}^k g_i^\star(\boldsymbol{x}) = 1$ and $\sum_{i=1}^k p(y = i \mid \boldsymbol{x}) = 1$, we have

$$\sum_{i=1}^k g_i^\star(\boldsymbol{x}) = \frac{1}{\lambda} \sum_{i=1}^k p(y = i \mid \boldsymbol{x}) = 1.$$

Therefore, we can easily obtain $\lambda = 1$. In this way, $g_i^\star = \frac{1}{\lambda} p(y = i \mid \boldsymbol{x}) = p(y = i \mid \boldsymbol{x})$, which concludes the proof.

**Mean Squared Error** If the mean squared error is used, we have the following optimization problem:

$$\phi(g) = \sum_{i=1}^{k}(p(y = i \mid \boldsymbol{x}) - g_i(\boldsymbol{x}))^2$$

$$\text{s.t. } \sum_{i=1}^{k} g_i(\boldsymbol{x}) = 1.$$

By using the Lagrange multiplier method, we can obtain the following non-constrained optimization problem:

$$\Phi(g) = \sum_{i=1}^{k}(p(y = i \mid \boldsymbol{x}) - g_i(\boldsymbol{x}))^2 + \lambda'(\sum_{i=1}^{k} g_i(\boldsymbol{x}) - 1)).$$

By setting the derivative to 0, we obtain

$$g_i^\star(\boldsymbol{x}) = p(y = i \mid \boldsymbol{x}) - \frac{\lambda'}{2}.$$

Because $\sum_{i=1}^{k} g_i^\star(\boldsymbol{x}) = 1$ and $\sum_{i=1}^{k} p(y = i \mid \boldsymbol{x}) = 1$, we have

$$\sum_{i=1}^{k} g_i^\star(\boldsymbol{x}) = \sum_{i=1}^{k} p(y = i \mid \boldsymbol{x}) - \frac{\lambda' k}{2}$$

$$0 = -\frac{\lambda' k}{2}.$$

Since $k \neq 0$, we can obtain $\lambda' = 0$. In this way, $g_i^\star = p(y = i \mid \boldsymbol{x}) - \frac{\lambda'}{2} = p(y = i \mid \boldsymbol{x})$, which concludes the proof.

### C.2 Proof of Theorem 5

According to Lemma 2, by minimizing $R_{cc}(f)$ with the cross entropy loss, we can obtain

$$q_j^\star(\boldsymbol{x}) = p(Y = C_j \mid \boldsymbol{x}), \forall j \in [2^k - 2].$$

Let us introduce $\widetilde{\boldsymbol{v}} = [p(Y = C_1 \mid \boldsymbol{x}), p(Y = C_2 \mid \boldsymbol{x}), \ldots, p(Y = C_{2^k - 2} \mid \boldsymbol{x})]$ and $\boldsymbol{v} = [p(y = 1 \mid \boldsymbol{x}), p(y = 2 \mid \boldsymbol{x}), \ldots, p(y = k \mid \boldsymbol{x})]$. We have

$$\widetilde{\boldsymbol{v}} = \boldsymbol{Q}^\top \boldsymbol{v}.$$

Since $q^\star(\boldsymbol{x}) = \widetilde{\boldsymbol{v}}$ and $g^\star(\boldsymbol{x}) = \boldsymbol{v}$, we have $q^\star(\boldsymbol{x}) = \boldsymbol{Q}^\top g^\star(\boldsymbol{x})$ where $g^\star(\boldsymbol{x}) = \text{softmax}(f^\star(\boldsymbol{x}))$. On the other hand, we can obtain $g_{cc}(\boldsymbol{x})$ by minimizing $R_{cc}$ (i.e., $g_{cc}(\boldsymbol{x}) = \text{softmax}(f_{cc}(\boldsymbol{x}))$), and thus $q^\star(\boldsymbol{x}) = \boldsymbol{Q}^\top g_{cc}(\boldsymbol{x})$, which further ensures $\boldsymbol{Q}^\top g^\star(\boldsymbol{x}) = \boldsymbol{Q}^\top g_{cc}(\boldsymbol{x})$. Therefore, when $\boldsymbol{Q}$ has full rank, we obtain $g_{cc} = g^\star$, which implies $f_{cc} = f^\star$. $\qquad \square$

## D   Proof of Theorem 6

Since this proof is somewhat similar to the proof of Theorem 4, we briefly sketch the key points.

We define a function space for our CC method as

$$\mathcal{G}_{cc} = \{(\boldsymbol{x}, Y) \mapsto \mathcal{L}(q(\boldsymbol{x}), \widetilde{y}) \mid f \in \mathcal{F}\},$$

where $(\boldsymbol{x}, Y)$ is randomly sampled from $\widetilde{p}(\boldsymbol{x}, Y)$ and $Y = C_{\widetilde{y}}$ (i.e., $Y$ is the $\widetilde{y}$-th label set in $\mathcal{C}$). Let $\widetilde{\mathfrak{R}}_n(\mathcal{G}_{cc})$ be the expected Rademacher complexity of $\mathcal{G}_{cc}$, i.e.,

$$\widetilde{\mathfrak{R}}_n(\mathcal{G}_{cc}) = \mathbb{E}_{\widetilde{p}(\boldsymbol{x}, Y)} \mathbb{E}_{\boldsymbol{\sigma}} \left[ \sup_{g \in \mathcal{G}_{cc}} \frac{1}{n} \sum_{i=1}^{n} \sigma_i g(\boldsymbol{x}_i, Y_i) \right].$$

Then we have the following lemma.

**Lemma 6.** *Suppose the loss function $\mathcal{L}$ is bounded by $M$, i.e., $M = \sup_{\boldsymbol{x} \in \mathcal{X}, f \in \mathcal{F}, \widetilde{y} \in \widetilde{\mathcal{Y}}} \mathcal{L}(q(\boldsymbol{x}), \widetilde{y})$, then for any $\delta > 0$, with probability at least $1 - \delta$,*

$$\sup_{f \in \mathcal{F}} \left| R_{\mathrm{cc}}(f) - \widehat{R}_{\mathrm{cc}}(f) \right| \leq 2 \widetilde{\mathfrak{R}}_n(\mathcal{G}_{\mathrm{cc}}) + \frac{M}{2} \sqrt{\frac{\log \frac{2}{\delta}}{2n}}.$$

*Proof.* In order to prove this lemma, we first show that the one direction $\sup_{f \in \mathcal{F}} R_{\mathrm{cc}}(f) - \widehat{R}_{\mathrm{cc}}(f)$ is bounded with probability at least $1 - \delta/2$, and the other direction can be similarly shown. Suppose an example $(\boldsymbol{x}_i, Y_i)$ is replaced by another arbitrary example $(\boldsymbol{x}_i', Y_i')$, then the change of $\sup_{f \in \mathcal{F}} R_{\mathrm{cc}}(f) - \widehat{R}_{\mathrm{cc}}(f)$ is no greater than $M/n$, since $\mathcal{L}$ is bounded by $M$. By applying *McDiarmid's inequality* [19], for any $\delta > 0$, with probability at least $1 - \delta/2$,

$$\sup_{f \in \mathcal{F}} R_{\mathrm{cc}}(f) - \widehat{R}_{\mathrm{cc}}(f) \leq \mathbb{E} \left[ \sup_{f \in \mathcal{F}} R_{\mathrm{cc}}(f) - \widehat{R}_{\mathrm{cc}}(f) \right] + M \sqrt{\frac{\log \frac{2}{\delta}}{2n}}.$$

Using the same trick in [20], we can obtain $\mathbb{E}[\sup_{f \in \mathcal{F}} R_{\mathrm{cc}}(f) - \widehat{R}_{\mathrm{cc}}(f)] \leq 2 \widetilde{\mathfrak{R}}_n(\mathcal{G}_{\mathrm{cc}})$. By further taking into account the other side $\sup_{f \in \mathcal{F}} \widehat{R}_{\mathrm{cc}}(f) - R_{\mathrm{cc}}(f)$, we have for any $\delta > 0$, with probability at least $1 - \delta$,

$$\sup_{f \in \mathcal{F}} \left| R_{\mathrm{cc}}(f) - \widehat{R}_{\mathrm{cc}}(f) \right| \leq 2 \widetilde{\mathfrak{R}}_n(\mathcal{G}_{\mathrm{cc}}) + M \sqrt{\frac{\log \frac{2}{\delta}}{2n}},$$

which concludes the proof. $\square$

Suppose the loss function $\mathcal{L}(q(\boldsymbol{x}), \widetilde{y})$ is $\rho'$-Lipschitz with respect to $f(\boldsymbol{x})$ ($0 \leq \rho \leq \infty$) for all $\widetilde{y} \in \widetilde{\mathcal{Y}}$, by the Rademacher vector contraction inequality [18], we can obtain $\widetilde{\mathfrak{R}}_n(\mathcal{G}_{\mathrm{cc}}) \leq \sqrt{2} \rho' \sum_{y=1}^{k} \mathfrak{R}_n(\mathcal{H}_y)$. By further taking into account Lemma 6 and Lemma 3, for any $\delta > 0$, with probability $1 - \delta$,

$$R_{\mathrm{cc}}(\widehat{f}_{\mathrm{cc}}) - R_{\mathrm{cc}}(f^\star) = R_{\mathrm{cc}}(\widehat{f}_{\mathrm{cc}}) - R_{\mathrm{cc}}(f_{\mathrm{cc}}) \leq 4 \sqrt{2} \rho' \sum_{y=1}^{k} \mathfrak{R}_n(\mathcal{H}_y) + 2M \sqrt{\frac{\log \frac{2}{\delta}}{2n}},$$

which concludes the proof of Theorem 6. $\square$

# E Detailed Information of Experiments

In this section, we provide more detailed information of the experiments.

## E.1 Datasets and Models

**Benchmark Datasets.** We use four widely-used benchmark datasets (including MNIST, Kuzushiji-MNIST, Fashion-MNIST, CIFAR-10) and five datasets (including Yeast, Texture, Dermatology, Har, 20Newsgroups) from the UCI Machine Learning Repository. The statistics of these datasets with the corresponding base models are reported in Table 5. It is worth noting that we only use the linear model on the UCI datasets, since they are not large-scale datasets. We report the descriptions of these datasets with the sources as follows.

- MNIST[1] [16]: It is a 10-class dataset of handwritten digits (0 to 9). Each instance is a 28×28 grayscale image.
- Kuzushiji-MNIST[2] [3]: It is a 10-class dataset of fashion items (T-shirt/top, trouser, pullover, dress, sandal, coat, shirt, sneaker, bag, and ankle boot). Each instance is a 28×28 grayscale image.

Table 5: Characteristics of the controlled datasets.

| Dataset | #Train | #Test | #Features | #Classes | Model |
|---------|--------|-------|-----------|----------|-------|
| Yeast | 1,335 | 149 | 8 | 10 | Linear Model |
| Texture | 4,950 | 550 | 40 | 11 | Linear Model |
| Dermatology | 329 | 37 | 34 | 6 | Linear Model |
| Har | 9,269 | 1,030 | 561 | 6 | Linear Model |
| 20Newsgroups | 16,961 | 1,885 | 300 | 20 | Linear Model |
| MNIST | 60,000 | 10,000 | 784 | 10 | three-layer ($d$-500-10) MLP, LeNet |
| Fashion-MNIST | 60,000 | 10,000 | 784 | 10 | three-layer ($d$-500-10) MLP, LeNet |
| Kuzushiji-MNIST | 60,000 | 10,000 | 784 | 10 | three-layer ($d$-500-10) MLP, LeNet |
| CIFAR-10 | 50,000 | 10,000 | 3,072 | 10 | 34-layer ResNet, 22-layer DenseNet |

Table 6: Characteristics of the real-world partially labeled datasets.

| Dataset | #Examples | #Features | #Classes | Avg. #CLs | Application Domain | Model |
|---------|-----------|-----------|----------|-----------|--------------------|-------|
| Lost | 1,122 | 108 | 16 | 2.23 | *automatic face naming* [21] | Linear Model |
| MSRCv2 | 1,758 | 48 | 23 | 3.16 | *object classification* [17] | Linear Model |
| BirdSong | 4,998 | 38 | 13 | 2.18 | *bird song classification* [2] | Linear Model |
| Soccer Player | 17,472 | 279 | 171 | 2.09 | *automatic face naming* [24] | Linear Model |
| Yahoo! News | 22,991 | 163 | 219 | 1.91 | *automatic face naming* [8] | Linear Model |

- Fashion-MNIST[3] [22]: It is a 10-class dataset of cursive Japanese ("Kuzushiji") characters. Each instance is a $28 \times 28$ grayscale image.

- CIFAR-10[4] [15]: It is a 10-class dataset of 10 different objects (airplane, bird, automobile, cat, deer, dog, frog, horse, ship, and truck). Each instance is a $32 \times 32 \times 3$ colored image in RGB format. This dataset is normalized with mean $(0.4914, 0.4822, 0.4465)$ and standard deviation $(0.247, 0.243, 0.261)$.

- 20Newsgroups[5]: It is a 20-class dataset of 20 different newsgroups (sci.crypt, sci.electronics, sci.med, sci.space, comp.graphics, comp.os.ms-windows.misc, comp.sys.ibm.pc.hardware, comp.sys.mac.hardware, comp.windows.x, rec.autos, rec.motorcycles, rec.sport.baseball, rec.sport.hockey, misc.forsale, talk.politics.misc, talk.politics.guns, talk.politics.mideast, talk.religion.misc, alt.atheism, soc.religion.christian). We obtained the tf-idf features, and applied TruncatedSVD [9] to reduce the dimension to 300. We randomly sample 90% of the examples from the whole dataset to construct the training set, and the rest 10% forms the test set.

- Yeast, Texture, Dermatology, Har[6]: They are all the datasets from the UCI Machine Learning Repository. Since they are all regular-scale datasets, we only apply linear model on them. For each dataset, we randomly sample 90% of the examples from the whole dataset to construct the training set, and the rest 10% forms the test set.

We run 5 trials on the four benchmark datasets and run 10 trials on the five UCI datasets, and record the mean accuracy with standard deviation. For the used models, the detailed information of the used 34-layer ResNet [10] and 22-layer DenseNet [11] can be found in the corresponding papers.

**Real-World Partially Labeled Datasets.** We also use five real-world partially labeled datasets[7], including Lost, BirdSong, MSRCv2, Soccer Player, Yahoo! News. Table 6 reports the characteristics of these real-world partially labeled datasets, including Lost [4], Birdsong [2], MSRCv2 [17], Soccer Player [24], Yahoo! News [8]. These real-world partially labeled datasets come from several application domains. Specifically, Lost, Soccer Player, and Yahoo! News are from *automatic face naming*, Birdsong is from *bird song classification*, and MSRCv2 is from *object classification*. For

Table 7: Transductive accuracy of each method using neural networks on benchmark datasets. ResNet is trained on CIFAR-10, and MLP is trained on the other three datasets.

| | MNIST | Kuzushiji-MNIST | Fashion-MNIST | CIFAR-10 |
|---|---|---|---|---|
| RC | **98.81±0.02%** | **97.45±0.06%** | **94.30±0.09%** | **87.48±0.44%** |
| CC | 98.77±0.06% | 97.31±0.05%• | 93.55±0.14%• | 86.15±0.26%• |
| GA | 96.72±0.11%• | 94.85±0.08%• | 87.34±0.10%• | 76.70±0.21%• |
| NN | 97.25±0.08%• | 93.91±0.06%• | 88.83±0.18%• | 74.31±0.35%• |
| Free | 88.38±0.51%• | 83.73±0.31%• | 82.77±0.61%• | 17.74±1.11%• |
| PC | 93.42±0.12%• | 88.26±0.10%• | 85.54±0.18%• | 46.93±2.35%• |
| Forward | 98.68±0.04%• | 96.89±0.07%• | 91.48±0.26%• | 78.72±1.32%• |
| EXP | 98.70±0.03% | 97.03±0.12%• | 92.60±0.05%• | 79.52±0.56%• |
| LOG | 98.75±0.06% | 97.18±0.06%• | 93.52±0.06%• | 85.96±0.45% |
| MAE | 98.63±0.05%• | 97.01±0.04%• | 92.02±0.08%• | 74.31±3.24%• |
| MSE | 97.35±0.24%• | 95.61±0.06%• | 90.53±0.12%• | 69.81±2.43%• |
| GCE | 97.15±0.03%• | 95.41±0.04%• | 90.80±0.16%• | 77.77±0.60%• |
| Phuber-CE | 95.59±0.30%• | 91.66±0.23%• | 88.65±0.12%• | 65.42±0.96%• |

Table 8: Transductive accuracy of each method using neural networks on benchmark datasets. DenseNet is trained on CIFAR-10, and LeNet is trained on the other three datasets.

| | MNIST | Kuzushiji-MNIST | Fashion-MNIST | CIFAR-10 ResNet |
|---|---|---|---|---|
| RC | **99.46±0.02%** | 98.69±0.03% | **94.32±0.07%** | **86.77±0.47%** |
| CC | 99.43±0.03% | **98.78±0.01%** | 94.31±0.17% | 85.38±0.16%• |
| GA | 95.58±0.02%• | 97.13±0.02%• | 89.33±0.03%• | 75.38±0.23%• |
| NN | 98.72±0.04%• | 96.99±0.06%• | 90.35±0.19%• | 75.12±0.25%• |
| Free | 79.98±2.03%• | 84.01±1.36%• | 75.03±3.95%• | 46.65±0.35%• |
| PC | 95.32±0.13%• | 90.80±0.12%• | 85.39±0.18%• | 55.68±2.30%• |
| Forward | 99.25±0.04%• | 98.72±0.06% | 92.77±0.23%• | 78.74±1.41%• |
| EXP | 99.27±0.01%• | 98.38±0.11%• | 93.23±0.04%• | 79.84±1.22%• |
| LOG | 99.38±0.09% | 98.75±0.06% | 93.52±0.07% | 84.10±0.54%• |
| MAE | 99.29±0.03%• | 98.47±0.17%• | 90.10±3.41%• | 74.05±0.87%• |
| MSE | 98.71±0.03%• | 95.53±0.17%• | 90.81±0.18%• | 79.12±0.40%• |
| GCE | 98.84±0.02%• | 97.48±0.16%• | 91.72±0.08%• | 79.47±0.38%• |
| Phuber-CE | 97.31±0.07%• | 92.44±1.19%• | 88.94±0.11%• | 70.73±0.39%• |

automatic face naming, each face cropped from an image or a video frame is taken as an instance, and the names appearing on the corresponding captions or subtitles are considered as candidate labels. For object classification, each image segment is regarded as an instance, and objects appearing in the same image are taken as candidate labels. For bird song classification, singing syllables of the birds are represented as instances and bird species jointly singing during a 10-seconds period are regarded as candidate labels. For each real-world partially labeled dataset, the average number of candidate labels (Avg. #CLs) per instance is also recorded in Table 6. In the experiments, we run 10 trials (with 90%/10% train/test split) on each real-world partially labeled dataset, and the mean accuracy with standard deviation is recorded for each method. Note that most of the existing parametric PLL methods adopt the linear model, hence we also apply linear model on these real-world partially labeled datasets for fair comparisons.

On all the above datasets, we take the average accuracy of the last ten epochs as the accuracy for each trial. All the experiments are conducted on NVIDIA Tesla V100 GPUs. Since our proposed methods are compatible with any stochastic optimizer, the time complexity of optimization could be in the linear order with respect to the number of data points.

## E.2 Compared Methods

The compared PLL methods are listed as follows.

- SURE [6]: It iteratively enlarges the confidence of the candidate label with the highest probability to be the correct label.

- CLPL [4]: It uses a convex formulation by using the one-versus-all strategy in the multi-class loss function.

- IPAL [25]: It is a non-parametric method that applies the label propagation strategy [27] to iteratively update the confidence of each candidate label.
- PLSVM [5]: It is a maximum margin-based method that differentiates candidate labels from non-candidate labels by maximizing the margin between them.
- PLECOC [26]: It adapts the Error-Correcting Output Codes method to deal with partially labeled examples in a disambiguation-free manner.
- PLKNN [12]: It adapts the widely-used $k$-nearest neighbors method to make predictions for partially labeled examples.

For all the above methods, their parameters are specified or searched according to the suggested parameter settings by respective papers. It is worth noting that since all the compared PLL methods use full batch size, we also use full batch size (with 2000 training epochs) for our proposed methods RC and CC, to keep fair comparisons.

Besides, we also compare with various complementary-label learning methods for two reasons: 1) By regarding each non-candidate label as a complementary label, we can transform the partially labeled dataset into complementarily labeled dataset, thus we can directly use complementary label methods. 2) Existing complementary-label learning methods can be applied to deal with large-scale datasets. The compared complementary-label learning methods are listed as follows.

- PC [13]: It utilizes the pairwise comparison strategy (with sigmoid loss) in the multi-class loss function to learn from complementarily labeled data.
- Forward [23]: It conducts forward correction by estimating the latent class transition probability matrix to learn from complementarily labeled data.
- Free, NN, GA [14]: These are three methods adapted from the same unbiased risk estimator for learning from complementarily labeled data. For the Free method, it minimizes the original empirical risk estimator. For the NN method, it corrects the negative term in the risk estimator using max operator. For the GA method, it uses a gradient ascent strategy to prevent from overfitting.
- MAE, MSE, GCE, Phuber-CE [7]: These are four methods that insert conventional bounded multi-class loss functions into the unbised risk estimator for learning with multiple complementary labels.
- EXP, LOG [7]: They are two methods for learning with multiple complementary labels. For these two methods, upper-bound surrogate loss functions are used in the derived empirical risk estimator [7].

Hyper-parameters for all the methods are selected so as to maximize the accuracy on a validation set, which is constructed by randomly sampling 10% of the training set.

## E.3 Transductive Analysis

Here, we provide additional experiments to investigate the transductive accuracy of each method, i.e., the training set is evaluated with true labels. Table 7 and Table 8 report the transductive accuracy of each method using different neural networks on benchmark datasets. As shown in the two tables, our proposed methods RC and CC still significantly outperform other compared methods in most cases. In addition, it is worth noting that the gap of transductive accuracy between RC and CC is not so significant. However, as shown before, the gap of test accuracy between RC and CC is quite significant. These observations further support our conjecture that the estimation error bound of RC is probably tighter than that of CC.

## E.4 Performance Curves

Here, we record the test accuracy at each training epoch to provide more detailed visualized results. To avoid the overcrowding of many curves in a single figure, we only use seven methods including RC, CC, GA, NN, Free, PC, and Forward. The linear model and the MLP model are trained on the benchmark datasets. Figure 1 reports the experimental results of the seven methods for different datasets and models. Dark colors show the mean accuracy of 5 trials and light colors show the standard deviation. As shown in Figure 1, our proposed PLL methods RC and CC still consistently outperform other compared methods, even when the simple linear model is used.

Figure 1: Experimental results of different methods for different datasets and models. Dark colors show the mean accuracy of 5 trials and light colors show the standard deviation.

# F   Experiments on Effectiveness of Generation Model

Here, we would like to test the performance of our methods under different data generation processes. As indicated before, our proposed PLL methods are based on the proposed data generation model. Therefore, we would like to investigate the influence of different generation models on our proposed methods. We use *entropy* to measure how well given candidate label sets match the proposed generation model. By this measure, we could know ahead of model training whether to apply our proposed methods or not on a specific dataset. We expect that the higher the entropy, the better the match, thus the better the performance of our proposed methods. To verify our conjecture, we generate various candidate labels sets by different generation models. It is worth noting that the average number of candidate labels (Avg. #CLs) per instance plays an important role in partially labeled datasets. Intuitively, the performance of PLL methods would generally be better if trained on the datasets with smaller Avg. #CLs. The Avg. #CLs of our generation model is 5. Therefore, to keep fair comparisons, the Avg. #CLs of other studied generation models is also kept as 5.

Figure 2: Heatmaps of different generation processes of candidate label sets.

In following experiments, we still focus on the case where the candidate label set is independent of the instance. We additionally introduce the *class transition matrix* (denoted by $\boldsymbol{T}$) for partially labeled data, where $T_{ij}$ describes the probability of the label $j$ being a candidate label given the true label $i$ for each instance. Intuitively, $T_{ii} = 1$ always holds since the true label is always a candidate label. In this way, we provide various formulations of the matrix $\boldsymbol{T}$ to instantiate different generation models.

The studied generation models are illustrated in Figure 2. As shown in Figure 2, we provide six cases of generation models, and each of them holds a value of entropy. The value of entropy is calculated by the following two steps: 1) The matrix $\boldsymbol{T}$ is normalized by $P_{ij} = T_{ij}/(\sum_j T_{ij})$, $\forall i, j \in [k]$. 2) The entropy of the case is calculated by $-\frac{1}{k} \sum_{i=1}^{k} \sum_{j=1}^{k} P_{ij} \log P_{ij}$. As in our proposed generation model, given the true label, other labels have the same probability to be a candidate label, our case achieves the maximum entropy (i.e., 2.257).

## Footnotes

[1]http://yann.lecun.com/exdb/mnist/

[2]https://github.com/rois-codh/kmnist

[3] https://github.com/zalandoresearch/fashion-mnist

[4] https://www.cs.toronto.edu/~kriz/cifar.html

[5] http://qwone.com/~jason/20Newsgroups/

[6] https://archive.ics.uci.edu/ml/datasets.php

[7] http://palm.seu.edu.cn/zhangml/Resources.htm#partial_data