[Reviews · NeurIPS 2020]

Review 1

Summary and Contributions: The authors propose a generation process for candidate labels when learning with partially labeled data. Additionally, they formulate two partially labeled data algorithms that they show to be provably consistent.

Strengths: The authors propose novel methods for partial label learning and provide theoretical justifications for their work. The experiments compare to several baselines and the authors run their methods on several datasets

Weaknesses: The paper is not clear and a bit difficult to follow. There are some typos and the methodology for the algorithms could be better explained.

Correctness: The empirical validation seems correct as the authors run multiple trials and report mean and standard deviations of their results.

Clarity: No, the paper is not very clear as there are some typos in the text and the methodology section 4 is not well explained.

Relation to Prior Work: While the authors present prior works in their paper, they do not do a good job of explaining how these works relate or differ from each other. They do motivate their work with limitations of existing methods.

Reproducibility: Yes

Additional Feedback: I think the paper should be better organized to assist with clarity. One suggestion is to have a related works section that compares existing methods and their limitations. Weakly supervised learning is used as an umbrella term to classify different learning methods in the introduction. This is somewhat confusing as weakly supervised learning is a separate research area and not an umbrella term for the different research areas listed. See works on data programming, adversarial label learning, snorkel and its variants. ========================================= I acknowledge that I read the rebuttal and thank the authors for providing explanations to the questions and concerns I had.


Review 2

Summary and Contributions: This paper handles the Partial-Label Learning problem where each training example has a set of candidate labels. Since most of the previous works do not consider the data distribution, there lacks a theoretical understanding of the consistency. Inspired by this, this paper proposes a novel statistical model to depict the generation process of partially labeled data (i.e. a set of candidate labels). Based on the generation model authors further develop consistent PLL methods, both risk-consistent and classifier-consistent verify the consistency. Authors theoretically derive an estimation error bound for each of the methods and show that the error bound of the risk-consistent method is tighter than the other method. In the experiment, the authors show the effectiveness of the method in the benchmark and real-world partially labeled dataset. == post review Thanks for the author's response. I understand the difference between PLL and NLL; the difference between them might be the ambiguity. Although the paper still has some unclear points as R1 review, it might be complemented in the final version. After the rebuttal process, I decided to increase my rating.

Strengths: If their statistical model is novel (please check the weaknesses first) + By depicting the explicit data distribution, this paper effectively defined the PLL problem as an empirical risk minimization. + The suggested two novel methods (i.e., risk-consistent and classifier-consistent) outperform the existing works on benchmark and real-world dataset.

Weaknesses: - Major: The proposed generation process assume that the correct label is always included in the set of candidate labels, and the set of candidate labels set is uniformly sampled (line 42 and 124). I do not know the difference between the suggested data distribution and a general noisy label setting [Chen, Pengfei, et al. "Understanding and Utilizing Deep Neural Networks Trained with Noisy Labels." ICML. 2019.] [Ren, Mengye, et al. "Learning to Reweight Examples for Robust Deep Learning." ICML. 2018.].

Correctness: Good

Clarity: Not significant

Relation to Prior Work: Somewhat weak

Reproducibility: Yes

Additional Feedback: Please check the weakness. If the authors address my concerns, I will re-consider my rating.


Review 3

Summary and Contributions: This paper studies the partial label learning problem by proposing a data generation process. The data generation process guarantee that the true label is contained in the label candidate set provided. The paper then proposed two methods for partial label learning. These methods are statistically consistent from two different directions, flexible with the choice on base model and with guaranteed performance. Finally, experimental results on the proposed method as well as the effectiveness of the proposed generation model are shown.

Strengths: The paper’s primary goal is to form a solid generation model for partial label learning. The paper successfully achieves this goal: it proposed a generation model agreeing well with the basic assumption on partial label learning, i.e., the true label needs to lie in the candidate label set. The generation model is also well-motivated by assigning equal weight to sample every subset containing the true label. As far as I know, there is no study on the generation model of partial label learning before, despite the fact that the generation model is important to generate the data and get the distribution of the data. In the part of the experiment, the paper also uses the entropy to test whether the given data agrees well with the generation model. This shows another value of the generation model: to help us know whether the proposed methods will work given some data. This result is really impressive, useful, and not encountered much in other machine learning literature. Based on the generation model, the paper further proposes two methods, both guaranteed but with slightly different statistical properties. Theoretically, given infinite data, both methods should achieve the optimal classifier as using ordinary labels. The paper analyses the methods through generalization error bound. The empirical results also agree with the theoretical results. Generally, the paper is well-organized, well-written. It studied an important problem for partial label learning, proposed novel generation model, and theoretical guaranteed method. I think the paper could make itself more self-consistent by giving more discussions on the difference between risk-consistent method and classifier-consistent method.

Weaknesses: After reading the paper, I am a bit confused about why we need a classifier-consistent method since both theoretically and empirically it is worse than the risk-consistent method. Is it because some literature focusing on proposing a classifier-consistent method, or there are some limitations on the loss functions used? I would suggest the paper to have more discussion on this part.

Correctness: yes

Clarity: yes

Relation to Prior Work: yes

Reproducibility: Yes

Additional Feedback: I have read the author response and peer's reviews. I think the paper contains some things interesting and novel. The finding also contains something important to the partial label learning. I would like to keep the rating.

[Author Response · NeurIPS 2020]

**Reviewer #1**

**Q1:** This paper is not clear as there are some typos in the text and the methodology section 4 is not well explained.

**A1:** Thank you very much for pointing out this issue. We will carefully revise and re-organize our paper to make it clearer and easier to understand. Here, we will clarify potential misunderstanding, if any, for our methods. For the methodology section (Section 4), we would try our best to offer additionally plain language explanations for the high-level idea of our proposed methods. However, as our motivation for deriving consistent methods based on the data generation model is purely theoretical, we need to give formal mathematical definition and derivations, which are important, necessary and may not be simply replaced by an easy-to-understand plain language explanation.

**Q2:** While the authors present prior works in their paper, they do not do a good job of explaining how these works relate or differ from each other. They do motivate their work with limitations of existing methods.

**A2:** Thank you for your comments. We agree that we motivated our work from existing methods, i.e., none of the existing methods considers a data generation process and the consistency of these methods would be hardly guaranteed. Our paper is the first work that solves the partial-label learning problem (with two novel provably consistent methods) through the lens of the data generation process. Previous works improve the practical performance of partial-label learning using various strategies, including the EM iterative procedure [17,22,28], maximum margin [31], manifold regularization [32,23], and error-correcting output codes [34]. Theoretical works [18,19] make the same assumption on ambiguity degree, with the difference that [18] provides a classifier-consistent method while [19] only focuses on learning theory. We will take your valuable suggestion to have a related work section that compares these methods and their limitations, and provide more detailed explanations in our final version.

**Reviewer #2**

**Q1:** I do not know the difference between the suggested data distribution and a general noisy label setting.

**A1:** There is an evident difference of the problem setting between *partial-label learning* (PLL) and *noisy-label learning* (NLL). Specifically, we denote by the data distribution for PLL $p(\boldsymbol{x}, Y)$ where $Y$ is a set of candidate labels and the data distribution for NLL $p(\boldsymbol{x}, \tilde{y})$ where $\tilde{y}$ is a single observed label that may not be the correct label. We can see that a set of candidate labels is generated by the former distribution, while a single observed label is generated by the latter distribution. Two different data forms cannot share the same data distribution. In our proposed data distribution for PLL, each possible candidate label set that contains the correct label will be uniformly sampled as the observed candidate label set. Since there are in total $2^{k-1} - 1$ possible candidate label sets, each candidate label set will be chosen to be the observed candidate label set with probability $1/(2^{k-1} - 1)$. For example, suppose the label space is $\{1, 2, 3, 4\}$ (i.e., $k = 4$) and the true label is 2 for a given instance, there are $2^{k-1} - 1 = 7$ possible candidate label sets: $\{1, 2, 3\}, \{1, 2, 4\}, \{2, 3, 4\}, \{1, 2\}, \{2, 3\}, \{2, 4\}, \{2\}$, each of them would be selected as the observed candidate label set with probability $1/7$. In contrast, in the data distribution (symmetric noise) for NLL, the correct label for a given instance has some probability (denoted by $z$) to be the observed label, and each of the other labels has a probability of $(1 - z)/(k - 1)$ to be the observed label. This concrete example also shows the difference between the two data distributions.

**Reviewer #3**

**Q1:** I am a bit confused about why we need a classifier-consistent method since it is worse than the risk-consistent method. Is it because some literature focusing on proposing a classifier-consistent method, or there are some limitations on the loss functions used? I would suggest the paper to have more discussion on this part.

**A1:** Thank you for your insightful comments. Yes, it is partly because some literatures focus on proposing a classifier-consistent method. We would like to show that based on our proposed data generation process, we can also derive a novel classifier-consistent method. Another reason is that in *noisy-label learning*, classifier-consistent methods are generally better than risk-consistent methods because risk-consistent methods usually overfit due to the negative empirical risk issue. In *partial-label learning* (PLL), we would like to provide the first study that compares the two types of consistent methods, which motivates us to also propose a classifier-consistent method. It is interesting that our risk-consistent PLL method does not have the overffiting issue since there is no negative empirical risk, hence it outperforms the classifier-consistent PLL method. In addition, we would like to emphasis the importance of the data generation process. We would like to show that having such data distribution, we are able to derive consistent (risk-consistent and classifier-consistent) methods. Both consistent methods can be considered as the products of the data generation process. We will provide more discussions on the difference between the risk-consistent method and the classifier-consistent method. There is no limitation on the used loss functions for the risk-consistent method. There is a small restriction (i.e., Lemma 3) on the used loss functions for the classifier-consistent method, while we have shown that common loss functions (e.g., the softmax cross entropy loss and mean squared error) can satisfy this condition. Intuitively, the performance of the two methods would be affected by the used loss functions. It would be interesting to investigate the influence of different loss functions on our methods in future work.

[Meta-Review · NeurIPS 2020]

This paper presents a method for learning with a set of candidate labels, under the assumption that all label sets that contain the true label are equally likely. This is an odd assumption, but results on 5 partial label learning datasets (that do not necessarily satisfy this assumption) look promising. The paper is detailed and the empirical results compare to 6 other methods.